# Increasing central and northern European summer heatwave intensity due to forced changes in internal variability

Goratz Beobide-Arsuaga [1] ✉, Laura Suarez-Gutierrez[2,3], Armineh Barkhordarian [1], Dirk Olonscheck [4] & Johanna Baehr [1]

In recent years, European summer heatwaves have strongly intensified due to rising anthropogenic emissions. While European summer heatwaves will continue to intensify due to the warming of summer temperatures, the effects of the changes in internal variability under global warming remain unknown. Employing five single-model initial-condition large ensembles, we find that the forced changes in internal variability are projected to intensify central and northern European summer heatwaves. Central and northern Europe will experience frequent moisture limitations, enhancing land-atmosphere feedback and increasing heatwave intensity and variability. In contrast, the forced changes in internal variability will contribute to weakening southern European summer heatwaves. Southern Europe is projected to face a more stable moisture-depleted environment that reduces extreme temperature variability and heatwave intensity. Our findings imply that while adaptation to increasing mean temperatures in southern Europe should suffice to reduce the vulnerability to increasing EuSHW intensity, in central and northern Europe adaptation to increased temperature variability will also be needed.

With human-induced rising global mean temperatures, the intensity, frequency, and duration of extreme heat events have increased almost everywhere since the 1950s, and this positive trend is expected to continue in the coming decades[1-5]. Europe, in particular, has been identified as a heatwave hotspot with increasing intensity trends up to four times higher than the rest of the northern midlatitudes[6,7]. During the last two decades, the occurrence of multiple unprecedented European summer heatwaves (EuSHWs) (e.g. 2003, 2015, 2018, and 2022) has caused large economic, ecological, and humanitarian losses[8-11].

The recent increase in EuSHW intensity has been partly explained by the forced shift in climatology towards warmer values[12-15]. Under rising global mean temperatures, European summer temperatures are also rising, altering the background mean state and shifting summer temperature distributions. Assuming a limited adaptation towards increasing European summer mean temperatures, the threshold at

which temperatures are perceived as extreme will be time-invariant, and consequently, the positive shift in the temperature distribution leads to an increase in EuSHW intensity[16]. For instance, considering an ecosystem that historically lived with average summer temperatures of 20 °C and extreme temperatures of 30 °C, a positive shift in temperature distributions of 5 °C will increase the mean to 25 °C and extreme temperatures to 35 °C. Although the difference between the shifted average summer temperatures and extreme temperatures has not changed from the historical period, the ecosystem will experience an increase in extreme heat if it does not adapt to the altered climatology.

In addition to the shift in the temperature distribution, the internal variability of the climate system has significantly contributed to the occurrence of recent heatwaves[17-20]. The variability in sea surface temperatures, soil moisture content, atmospheric dynamics, and

[1]Institute of Oceanography, Center for Earth System Research and Sustainability (CEN), Universität Hamburg, Hamburg, Germany. [2]Institute for Atmospheric and Climate Science, ETH Zurich, Zurich, Switzerland. [3]Laboratoire des Sciences du Climat et de l'Environnement, Institut Pierre-Simon Laplace, Paris, France. [4]Max Planck Institute for Meteorology, Hamburg, Germany. ✉e-mail: goratz.beobide.arsuaga@uni-hamburg.de

the consequent positive feedbacks involved have been related to historical EuSHWs[6,21–27]. Persistent sea surface temperature anomalies in the North Atlantic, Pacific and Barents Sea have been critical to sustaining circumglobal atmospheric wave trains and consequent long-lasting high-pressure systems over Europe, which relate to negative precipitation and soil moisture anomalies[17,19,20,24,25]. In transitional regimes, a decrease in soil moisture reduces evaporative cooling by reducing latent heat flux and increases temperatures by increasing sensible heat flux, feeding back onto the atmospheric circulation, further decreasing soil moisture content and amplifying the intensity of recent EuSHWs[23,25].

However, under rising global mean temperatures, the background mean warming is expected to alter the mean state of the ocean, land, atmosphere and their interactions, and therefore, the variability of the climate system[15,21,28–35]. In the upcoming decades Europe is expected to face significant drying[36,37], which, depending on the historical soil moisture mean state, could change regions from energy-limited regimes into transitional regimes, and from transitional regimes into dry regimes (Fig. 5 in ref. 23). Considering that soil moisture-temperature feedback is effective only in transitional regimes, a decrease in soil moisture content could lead to either enhanced land-atmosphere coupling and temperature variability, or diminished land-atmosphere coupling and temperature variability. An increasing temperature variability could enhance the increase in the heatwave intensity caused by the shift in temperature distributions and expand the intensity range of possible future EuSHWs increasing the risk of unprecedented EuSHWs[38,39]. Following the previous example of the ecosystem, if the forced changes in internal variability induce an additional 2 °C increase in extreme temperatures, the difference between the shifted average summer temperatures and extreme temperatures will increase relative to historical times. Therefore, even if the ecosystem adapts to increasing mean temperatures and mitigates the effects of increasing heatwave intensity due to the distribution shift, this might not be enough to reduce the vulnerability of upcoming EuSHWs.

While it is well established in the literature that EuSHW intensity will continue to increase due to the shift in summer temperature distributions, how the forced changes in internal climate variability will affect EuSHW intensity under increasing global warming levels remains unknown. Previous studies have been limited by the short observational record that did not allow a robust sampling of EuSHWs, or by the coarse temporal resolution of climate models that did not allow the identification of heatwave events but rather an assessment of summer mean temperature metrics. Here, we employ five single model initial condition large ensembles (SMILEs): ACCESS-ESM-1.5[40] (40 members), CanESM5[41] (50 members), EC-Earth3[42] (50 members), MIROC6[43] (50 members), and MPI-GE CMIP6[44] (50 members). We use all available SMILEs that provide ~50 ensemble members with daily temporal resolution for the late historical period (1970–2014) and two shared socioeconomic pathway scenarios (SSP2–4.5, and SSP5–8.5) that span until the end of the 21st century. The high temporal resolution enables us to identify heatwave events and quantify their intensity, and the ~50 ensemble members for each SMILE enable us to robustly differentiate the effects of changing internal climate variability (i.e. forced changes in internal variability) from shifting temperature distributions (i.e. forced distribution shift)[45–47]. Here, we investigate the changes in EuSHW intensity and their range due to forced changes in internal variability and forced distribution shift under different global warming levels.

## Results

### Historical and projected European summer heatwave intensity
The intensity of non-detrended EuSHWs (see methods for details) has been increasing over the past two decades leading to the three most extreme observed events in 2022, 2003, and 2015, in the respective

order, and the increase is projected to continue until ~2040 regardless of the SSP scenarios for all climate models (Fig. 1a, d, g, j, m). Thereafter, the forced signal in EuSHW intensity, computed as the ensemble mean, and the range of possible future EuSHW intensity due to the internal climate variability will strongly depend on the SSP scenarios. The five climate models indicate that for the SSP2–4.5 scenario the increase in EuSHW intensity will be relatively constant, and for the SSP5–8.5 scenario the increase will be non-linear. By the end of the century, CanESM5 shows the highest forced signal in EuSHW intensity (25 times higher than the mean observed EuSHW intensity for SSP2–4.5, and 108 times higher for SSP5–8.5; Fig. 1d), followed by ACCESS-ESM-1.5 (24 times for SSP2–4.5, and 76 times for SSP5–8.5; Fig. 1a), EC-Earth3 (18 times for SSP2–4.5, and 76 times for SSP5–8.5; Fig. 1g), MIROC6 (14 times for SSP2–4.5, and 57 times for SSP5–8.5; Fig. 1j) and MPI-GE CMIP6 (9 times for SSP2–4.5, and 47 times for SSP5–8.5; Fig. 1m), in the respective order.

The range of possible future EuSHW intensity due to the internal climate variability computed as the difference between the 95th and the 5th ensemble percentile follows a similar evolution (Fig. 1a, d, g, j, m). For each model, the range increases similarly in both forcing scenarios until ~2040. However, the range of EuSHWs due to internal climate variability will diverge towards the end of the 21st century with the largest increase shown by the SSP5–8.5 scenario. ACCESS-ESM-1.5 shows the largest range that is 59 times larger than the mean observed EuSHW intensity for SSP5–8.5 (Fig. 1a), followed by MPI-GE CMIP6 (50 times larger; Fig. 1m), EC-Earth3 (47 times larger; Fig. 1g), CanESM5 (41 times larger; Fig. 1d), and MIROC6 (31 times larger; Fig. 1j). The qualitative time evolution of the forced signal and the range of possible EuSHW intensity due to internal climate variability reflects the evolution of the underlying radiative forcing, and hence, the evolution of global mean temperature anomalies.

### Sensitivity of EuSHW intensity to global warming levels
We find a robust relationship between global warming levels and the forced signal in five climate models (Fig. 1b, e, h, k, n), as expected, as well as a robust relationship between global warming levels and the range of possible EuSHW intensity (Fig. 1c, f, i, l, o). We find that on the one hand, the forced signal in EuSHW intensity increases non-linearly under increasing global mean temperature anomalies. On the other hand, the range of possible EuSHWs computed as ensemble standard deviation increases linearly under increasing global mean temperature anomalies. MPI-GE CMIP6 shows the largest increase in the range of EuSHW intensity under global warming (an increase of 3.9 times the mean observed EuSHW intensity per one degree Celsius increase of global mean temperature anomalies; Fig. 1o), followed by ACCESS-ESM-1.5 (3.5 times; Fig. 1c), MIROC6 (2.8 times; Fig. 1l), EC-Earth3 (2.7 times; Fig. 1i), and CanESM5 (2 times; Fig. 1f).

The scaling of the forced signal and the range of EuSHW intensity due to internal variability with global warming anomalies are not spatially uniform (Fig. 2). Although the increase in the forced signal (Fig. 2a) and in the range of EuSHW intensity (Fig. 2b) are projected for entire Europe, the strongest increase is projected for southern Europe and the increase will gradually reduce northward. All models agree on the positive relationship between the forced signal and the range of EuSHW intensity, and global warming anomalies for the entire European domain (Supplementary Fig. 1). Yet, models differ on the magnitude of the increase. MIROC6 shows the largest increase across models in the forced signal and in the range of EuSHW intensity in southern Europe, but it also shows the most abrupt reduction of the increase northward (Supplementary Fig. 1g, h). In contrast, CanESM5 shows the least abrupt reduction of the increase northward (Supplementary Fig. 1c, d), with ACCESS-ESM-1.5 (Supplementary Fig. 1a, b), MPI-GE CMIP6 (Supplementary Fig. 1i, j) and EC-Earth3 (Supplementary Fig. 1e, f) in between.

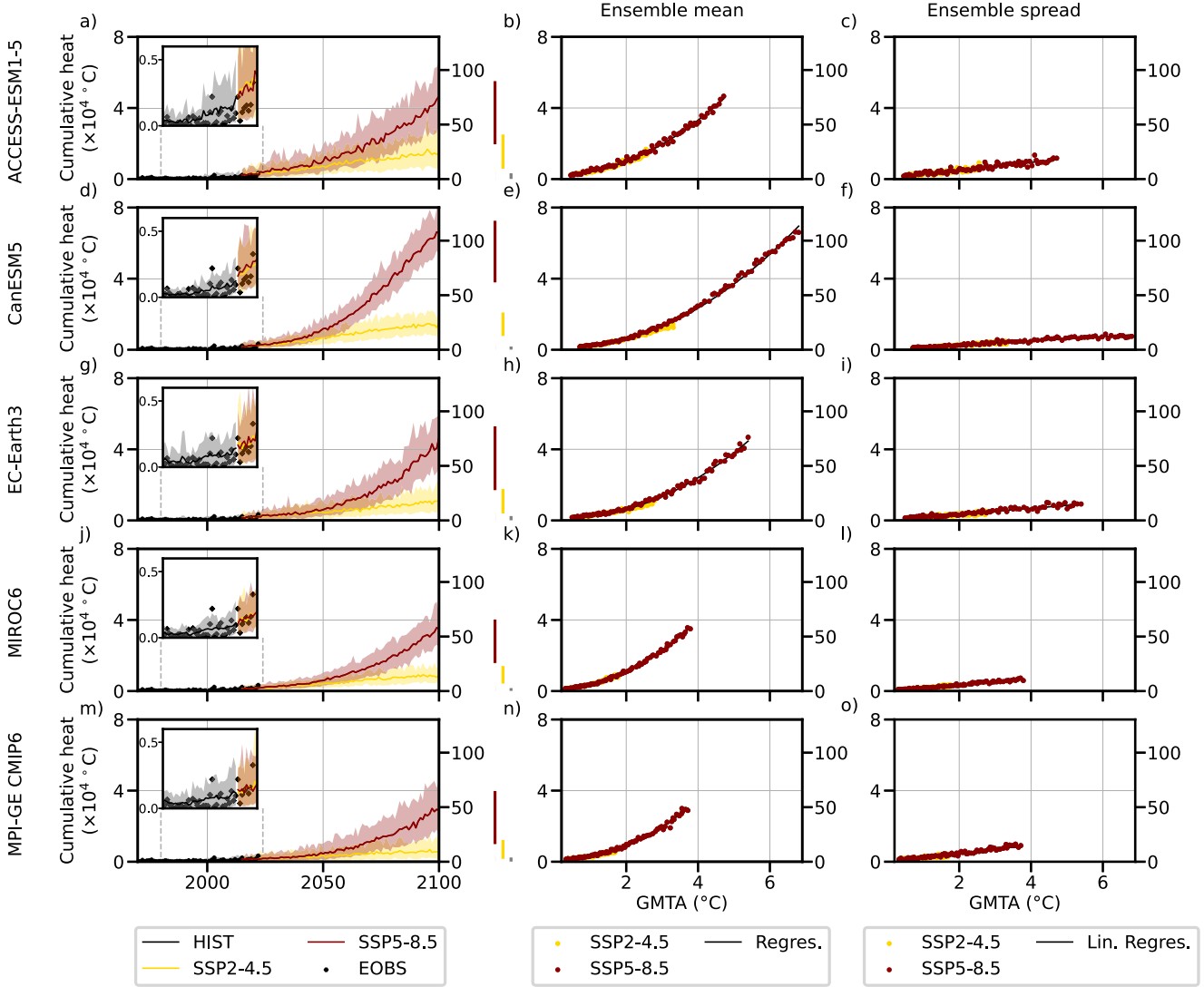

**Fig. 1 | Changes in European summer heatwave (EuSHW) intensity. a** Observed, historical and projected non-detrended EuSHW intensity expressed as cumulative heat in the left y-axis and scaled it to the observed historical mean (1970–2022) in the right y-axis. Solid lines indicate the ensemble mean, the shading indicates the 5th and 95th percentile, and the right vertical bars indicate the 5th and 95th percentile for the last 20 years of the historical simulation and each shared socioeconomic pathway (SSP) scenario for ACCESS-ESM-1.5; **b** the forced signal (i.e., ensemble mean) of non-detrended EuSHW cumulative heat scaled to global mean temperature anomalies (GMTA) relative to 1985–2014 for SSP2–4.5 and SSP5–8.5 scenarios (2014–2100) for ACCESS-ESM-1.5; **c** the range (i.e., ensemble spread computed as ensemble standard deviation) of non-detrended EuSHW cumulative heat due to internal variability scaled to GMTA relative to 1985–2014 for SSP2–4.5 and SSP5–8.5 scenarios (2014–2100) for ACCESS-ESM-1.5; **d–f** same as (**a–c**) but for CanESM5; **g–i** same as (**a–c**) but for EC-Earth3; **j–l** same as (**a–c**) but for MIROC6; **m–o** same as (**a–c**) but for MPI-GE CMIP6.

The increase in the forced signal in non-detrended EuSHW intensity and its range is driven by the combined effects of the forced distribution shift and the forced changes in internal variability. The forced distribution shift implies that temperature distributions will shift towards positive values and thus more frequent exceeding the fixed threshold. With a fixed percentile-based heatwave threshold (i.e. assuming a constant perception of what extreme temperatures are for a given day of the summer and grid point) a positive shift in temperature distribution will lead to an increased EuSHW intensity together with an increase in their possible range, even if the internal variability remains unforced. The forced changes in internal variability imply a change in the variability of the climate system and hence, of temperatures, which could enhance or suppress the forced signal and the range of EuSHW intensity caused by the shift in temperature distributions. We next disentangle the contribution of the forced changes in internal variability from the forced distribution shift.

## The effects of the forced changes in internal variability

We detrend daily maximum 2 m air temperature by subtracting the ensemble mean to remove the effects of the forced distribution shift, and focus on the effects of forced changes in internal variability (Fig. 3, see methods for details). The recent and projected increase in non-detrended EuSHW intensity seen in Fig. 1 is considerably reduced when removing the effect of mean temperature rise, which implies that the forced distribution shift is the main contributor to the increase in EuSHW intensity (Fig. 3a, d, g, j, m). Yet, we find a positive linear relationship between global warming levels and the forced signal in all models (Fig. 3b, e, k, n), excluding EC-Earth3 which shows a slightly negative linear relationship (Fig. 3h). The linear relationship between global warming levels and the range of detrended EuSHW intensity is consistently positive in all models (Fig. 3c, f, i, l, o). The positive linear relationship implies that the forced changes in internal variability will enhance the intensity and variability of EuSHWs caused by the temperature distribution shift, increasing the risk of unprecedented

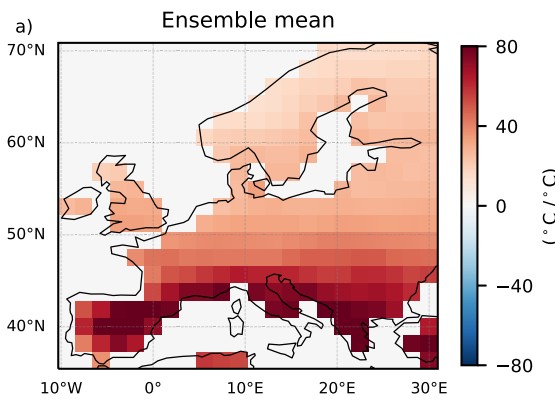
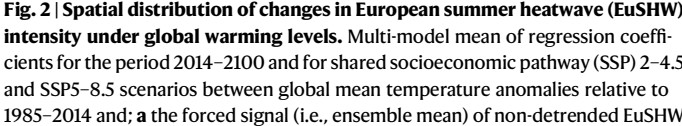
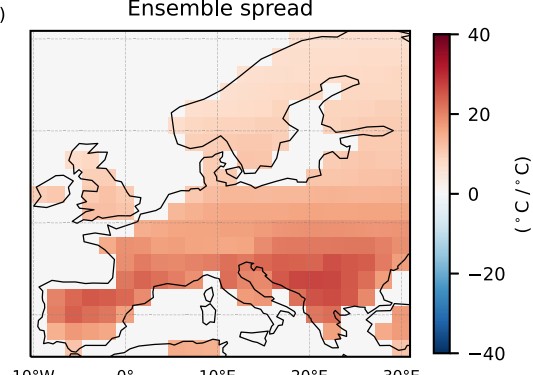

**Fig. 2 | Spatial distribution of changes in European summer heatwave (EuSHW) intensity under global warming levels.** Multi-model mean of regression coefficients for the period 2014–2100 and for shared socioeconomic pathway (SSP) 2–4.5 and SSP5–8.5 scenarios between global mean temperature anomalies relative to 1985–2014 and; **a** the forced signal (i.e., ensemble mean) of non-detrended EuSHW cumulative heat; **b** the range (i.e., ensemble standard deviation) of non-detrended EuSHW cumulative heat due to internal variability. In the non-dashed regions at least three out of the five models agree on the regression coefficient sign, and at least three out of the five models show significant changes at the 95% confidence level.

EuSHWs. MPI-GE CMIP6 shows the largest increase in the forced signal and the range of EuSHW intensity across models (an increase of 0.19 and 0.22 times the mean observed EuSHW intensity per one degree Celsius increase of global mean temperature anomalies, respectively; Fig. 3n, o), followed by ACCESS-ESM-1.5 (0.18 and 0.12 times; Fig. 3b, c), MIROC6 (0.9 and 0.11 times; Fig. 3k, l), CanESM5 (0.02 and 0.04 times; Fig. 3e, f), and EC-Earth3 (a decrease of 0.01 and an increase of 0.01 times; Fig. 3e, f).

We find a robust increase in the forced signal and the range of EuSHW intensity with global warming levels emerging from central and northern Europe (Fig. 4a, b). Considering that individual models largely disagree on the regional patterns (Supplementary Fig. 2), we define a robust change in the forced signal and the range of EuSHW intensity (non-dashed regions in Fig. 4a, b) when at least three out of the five models agree on the sign of the change and three out of the five models show significant changes at the 95% confidence level. All models except MIROC6 show a significant increase in the forced signal and the range of EuSHW intensity in central Europe, although in ACCESS-ESM-1.5 this increase is more pronounced in northern than in central Europe. Similarly, all models except CanESM5 and EC-Earth3 show a significant increase in northern Europe (Supplementary Fig. 2). The maximum positive contribution of the forced changes in internal variability to the increase in the forced signal and the range of central-northern EuSHW intensity (45°N latitude northward) is shown by ACCESS-ESM-1.5 (14 % and 32%, respectively; Supplementary Fig. 3a, b), followed by MPI-GE CMIP6 (14 % and 24%; Supplementary Fig. 3i, j), MIROC6 (7% and 11%; Supplementary Fig. 3g, h), EC-Earth3 (2% and 8%; Supplementary Fig. 3e, f), and CanESM5 (1% and 6%, respectively). In central and northern Europe the forced changes in internal variability will cause a faster increase of the right tail of daily maximum temperatures than the mean under increasing global warming levels (Supplementary Fig. 4), that is, an increase in temperature variability in agreement with ref. 21 and ref. 34. A faster increase of extreme summer temperatures compared to the mean leads to the positive trend of the forced signal in EuSHW intensity after detrending daily maximum temperatures. In addition, the increased temperature variability due to the forced changes in internal variability will lead to a higher fluctuation of extreme temperatures (Supplementary Fig. 5) leading to a larger range of EuSHW intensity.

In contrast, we find a robust decrease in the forced signal and the range of EuSHWs with global warming levels along the Mediterranean coastline (Fig. 4a, b). While CanESM5 and MIROC6 show a decrease confined to the southern Iberian Peninsula, Italy and Greece, ACCESS-ESM-1.5, EC-Earth3 and MPI-GE CMIP6 show also a decrease in the Mediterranean coast and the Balkans (Supplementary Fig. 2). The maximum negative contribution of the forced changes in internal variability to the increase in the forced signal and the range of southern EuSHW intensity (45°N latitude southward) is shown by ACCESS-ESM-1.5 (1% and 7%, respectively; Supplementary Fig. 3a, b), followed by EC-Earth3, MIROC6 and MPI-GE CMIP6 (equal contribution of 1% and 6%; Supplementary Fig. 3e–j), and CanESM5 (0.5% and 4%; Supplementary Fig. 3c, d). In southern Europe the forced changes in internal variability will cause a slower increase of the right tail of daily maximum temperatures than mean temperatures under global warming levels (Supplementary Fig. 4), that is, a decrease in temperature variability in agreement with ref. 48. A slower increase of extreme temperatures compared to the mean leads to a negative trend of the forced signal in EuSHW intensity after detrending daily maximum temperatures. Furthermore, the decreased temperature variability due to the forced changes in internal variability will lead to a smaller fluctuation of extreme temperatures (Supplementary Fig. 5) leading to a reduced range of EuSHW intensity.

Our results imply that while large model uncertainties exist, the forced changes in internal variability will enhance the increase in central and northern EuSHW caused by the forced distribution shift increasing the risk of unprecedented central and northern EuSHWs. In contrast, the forced changes in internal variability will partly mitigate the increase in southern EuSHW intensity caused by the forced distribution shift decreasing the risk of unprecedented southern EuSHWs. Therefore, while an adaptation to increasing mean daily maximum temperatures in southern Europe should be enough to reduce the vulnerability to upcoming EuSHWs, in central and northern Europe an adaptation to increased variability will also be needed.

### Relating the changes in EuSHW intensity to the changes in soil moisture

The regional differences in the effects of the forced distribution shift and the forced changes in internal variability on EuSHW intensity are related to the regional differences in soil moisture changes. We find a robust decrease in summer soil moisture content with increasing global warming levels in all Europe excluding some areas southeast of the Baltic Sea (Fig. 5a) which can be related to the robust projected increase in non-detrended EuSHW intensity (Fig. 2). A decrease in soil moisture would generally reduce the cooling effect of latent heat-flux and increase the heating effect of sensible heat-flux[49,50], leading to the overall increase in EuSHW intensity.

Under global warming, mean summer soil moisture declines continent-wide (Fig. 5a), but its impact on interannual variability differs

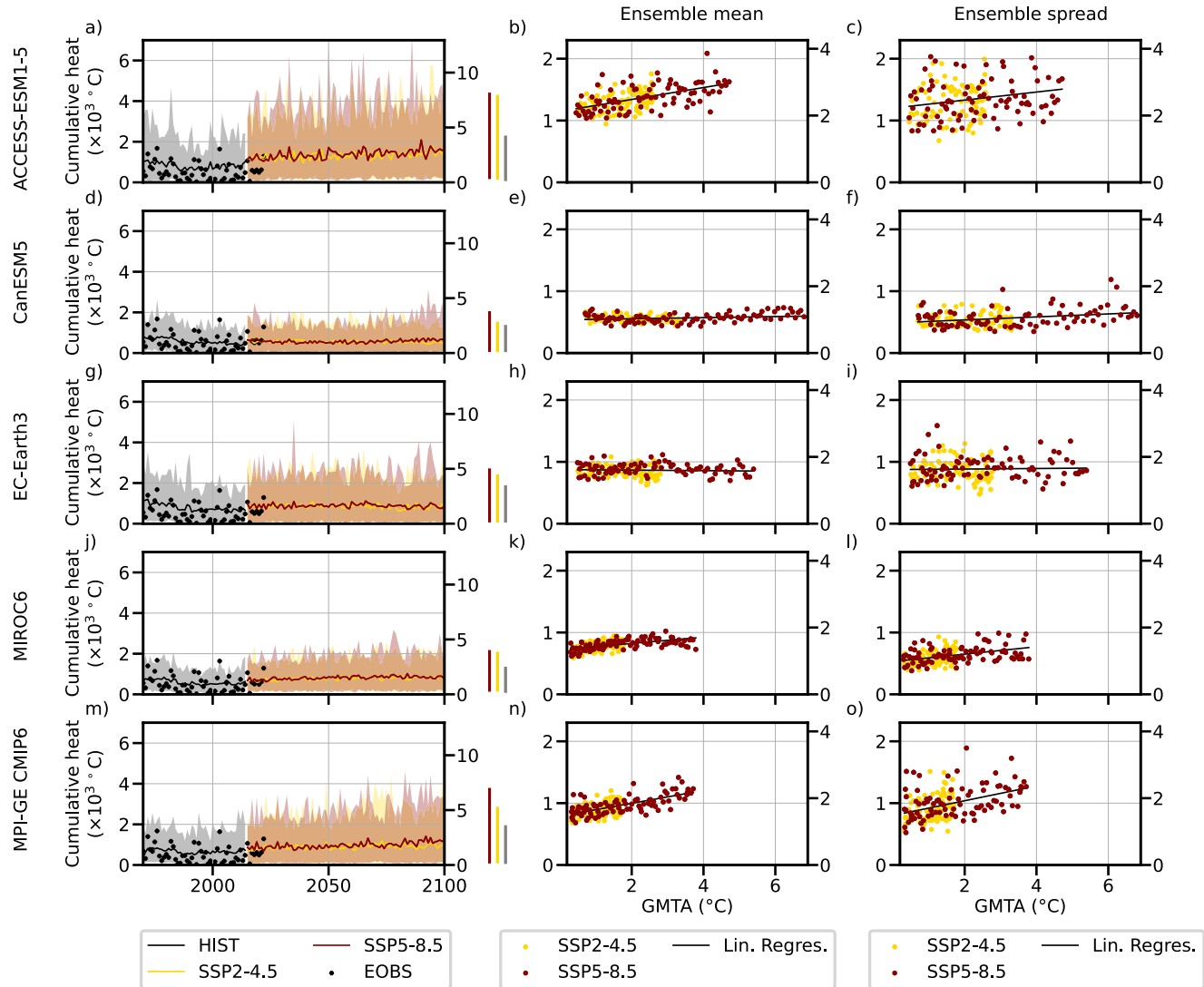

**Fig. 3 | Changes in European summer heatwave (EuSHW) intensity due to forced changes in internal variability. a** Detrended observed, historical and projected EuSHW intensity expressed as cumulative heat in the left y-axis and scaled it to the observed historical mean (1970–2022) in the right y-axis. Solid lines indicate the ensemble mean, the shading indicates the 5th and 95th percentile, and the right vertical bars indicate the 5th and 95th percentile for the last 20 years of the historical simulation and each shared socioeconomic pathway (SSP) scenario for ACCESS-ESM-1.5; **b** the forced signal (i.e., ensemble mean) of detrended EuSHW cumulative heat scaled to global mean temperature anomalies (GMTA) relative to 1985–2014 for SSP2–4.5 and SSP5–8.5 scenarios (2014–2100) for ACCESS-ESM-1.5; **c** the range (i.e., ensemble spread computed as ensemble standard deviation) of detrended EuSHW cumulative heat due to internal variability scaled to GMTA relative to 1985–2014 for SSP2–4.5 and SSP5–8.5 scenarios (2014–2100) for ACCESS-ESM-1.5; **d–f** same as (**a–c**) but for CanESM5; **g–i** same as (**a–c**) but for EC-Earth3; **j–l** same as (**a–c**) but for MIROC6; **m–o** same as (**a–c**) but for MPI-GE CMIP6.

by region (Fig. 5b). We find a robust increase in soil moisture variability in central and northern Europe, where we also find a robust increase in EuSHW intensity and its range due to the forced changes in internal variability (Fig. 4). We also find a robust decrease in soil moisture variability in the southernmost European grid-points coherent with the areas of decreased EuSHW intensity and range due to forced changes in internal variability. Soil moisture in southern Europe is historically lower than in central and northern Europe[51]. A further decrease in soil moisture over southern Europe will lead to a more constant state of moisture depletion state, which means less often switching between moisture limited and energy limited states, a weaker coupling between soil moisture and extreme summer daily maximum 2 m air temperature (Fig. 6a), and less effective land-atmosphere feedback (Fig. 6b), mitigating EuSHW intensity and its range due to the forced changes in internal variability. In contrast, a decrease in soil moisture over central and northern Europe will push the region to switch more often between moisture limited and energy limited states, resulting in a

stronger coupling between soil moisture and extreme summer daily maximum 2 m air temperature (Fig. 6a), an enhancement of land-atmosphere feedback (Fig. 6b), an increase in soil moisture variability (Fig. 5b), and an increase in the amplitude and variability of EuSHW intensity due to the forced changes in internal variability.

The inter-model differences in EuSHW intensity change due to the forced changes in internal variability (Supplementary Fig. 2) are partly related to the inter-model differences in soil moisture variability changes (Supplementary Fig. 6b, d, f, h). We find model disagreements on the sign of soil moisture variability change with global warming levels in eastern Europe, where we also find matching model disagreements on the sign of EuSHW intensity changes due to the forced changes in internal variability. ACCESS-ESM-1.5 and MIRO6 project a decrease in soil moisture variability and a decrease in EuSHW intensity and its range due to the forced changes in internal variability, while CanESM5, EC-Earth3 and MPI-GE CMIP6 project an increase in soil moisture variability and in EuSHW intensity and its range due to the

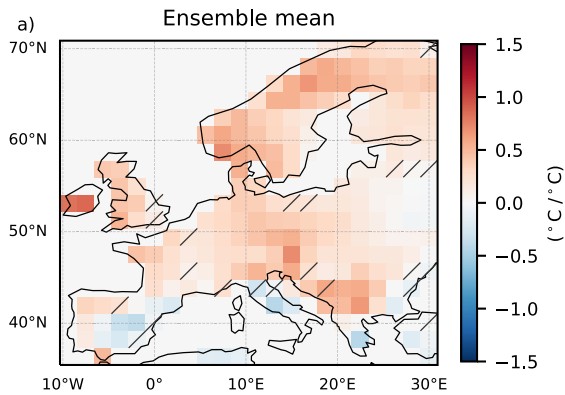
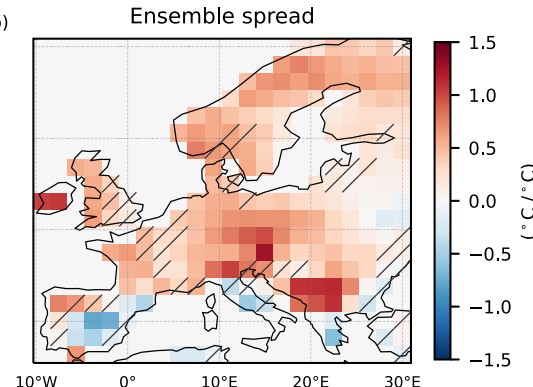

**Fig. 4 | Spatial distribution of changes in European summer heatwave (EuSHW) intensity due to forced changes in internal variability under global warming levels.** Multi-model mean of regression coefficients for the period 2014–2100 and for shared socioeconomic pathway (SSP) 2–4.5 and SSP5–8.5 scenarios between global mean temperature anomalies relative to 1985–2014 and; **a** the forced signal (i.e., ensemble mean) of detrended EuSHW cumulative heat; **b** the range (i.e., ensemble spread computed as ensemble standard deviation) of detrended EuSHW cumulative heat due to internal variability. In the non-dashed regions at least three out of the five models agree on the regression coefficient sign, and at least three out of the five models show significant changes at the 95% confidence level.

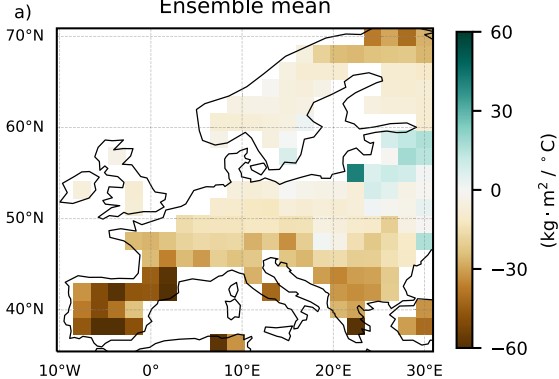
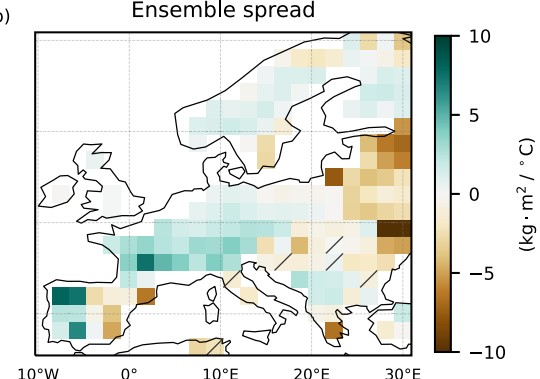

**Fig. 5 | Spatial distribution of changes in soil moisture under global warming levels.** Multi-model mean of regression coefficients for the period 2014–2100 and for shared socioeconomic pathway (SSP) 2–4.5 and SSP5–8.5 scenarios between global mean temperature anomalies relative to 1985–2014 and; **a** the forced signal (i.e., ensemble mean) of summer (June, July, August) mean soil moisture; **b** the range (i.e., ensemble spread computed as ensemble standard deviation) of summer mean soil moisture. In the non-dashed regions at least three out of the five models agree on the regression coefficient sign, and at least three out of the five models show significant changes at the 95% confidence level.

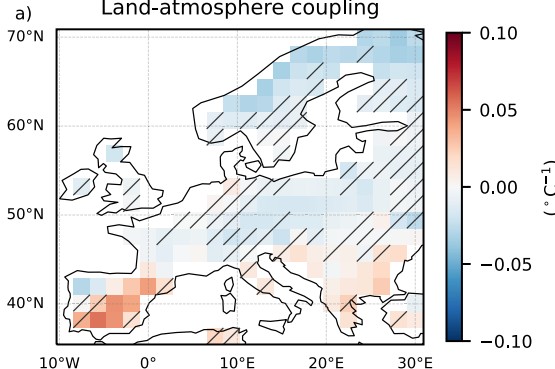
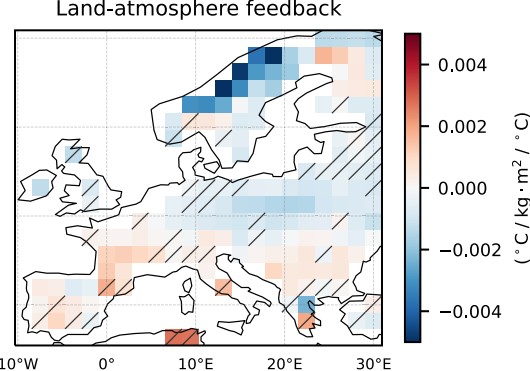

**Fig. 6 | Spatial distribution of changes in land-atmospheric coupling and feedback under global warming levels.** Multi-model mean of regression coefficients for the period 2014–2100 and for shared socioeconomic pathway (SSP) 2–4.5 and SSP5–8.5 scenarios between global mean temperature anomalies relative to 1985–2014 and; **a** land-atmospheric coupling computed as the correlation between 90th percentile summer (June, July, August) daily maximum 2 m air temperatures and summer mean soil moisture; **b** land-atmospheric feedback computed as the linear regression between 90th percentile summer daily maximum 2 m air temperatures and summer mean soil moisture. In the non-dashed regions at least three out of the five models agree on the regression coefficient sign, and at least three out of the five models show significant changes at the 95% confidence level.

forced changes in internal variability (Supplementary Fig. 2 and 6b, d, f, h, j). Yet, we also find EuSHW intensity changes that can not be explained with soil moisture changes, mainly in northern Europe. The decreased intensity simulated by CanESM5 north of the Baltic Sea (Supplementary Fig. 2c, d) can not be explained by the decrease in soil moisture (Supplementary Fig. 6c) and the increase in variability (Supplementary Fig. 6d) under global warming. Similarly, the increased intensity simulated by MPI-GE CMIP6 in northern Europe (Supplementary Fig. 2i, j) can not be explained by the increase in soil moisture (Supplementary Fig. 6i) and decrease in variability (Supplementary Fig. 6j) under global warming. It is likely that other factors (e.g., changes in the North Atlantic sea surface temperature or atmospheric variability) also contribute to the forced changes in internal variability, and hence affect northern EuSHW intensity.

## Discussion

Anthropogenic forcing will cause a non-linear increase in EuSHW intensity and a linear increase in their intensity range with increasing global warming anomalies, leading to larger EuSHW intensity fluctuations and risk of unprecedented EuSHWs. The main contributor to the increase in EuSHW intensity and the range is the positive shift in summer temperature distributions, which will increase the intensity and range in the entirety of Europe with the strongest impact in the south, in agreement with ref. 12 and ref. 13. Assuming limited adaptation to rising European summer temperatures, the threshold at which temperatures are perceived as extreme will be time-invariant. As a result, a positive shift in the summer temperature distribution will cause more frequent exceedances of the heatwave threshold, leading to an increase in both the intensity and range of EuSHWs, and amplifying the apparent role of internal variability in heatwaves, even if the internal variability itself remains unforced.

In addition, the forced changes in internal variability will enhance the increase in central and northern EuSHW intensity and its range. In central and northern Europe extreme maximum temperatures are projected to increase faster than the mean, which means that temperature variability will increase, enhancing the increase in EuSHW intensity due to the forced shift in temperature distribution. Furthermore, the variability of extreme maximum temperatures is projected to increase, which will increase the range of central and northern EuSHW intensity. In contrast, the forced changes in internal variability will partly mitigate the increase in southern EuSHWs. In southern Europe extreme maximum temperatures are projected to increase slower than the mean, which means that temperature variability will decrease, diminishing the increase in EuSHW intensity due to the forced shift in temperature distribution. Furthermore, the variability of extreme maximum temperatures is projected to decrease, which will decrease the range of southern EuSHW intensity.

The opposing effects of the forced changes in internal variability in central-north and southern EuSHWs can be explained by the non-linear effects of decreasing soil moisture. In central and northern Europe a decrease in soil moisture will cause the region to switch more often between moisture limited and energy limited states, amplifying land-atmosphere coupling and feedback, increasing temperature variability, and enhancing the intensity and variability of EuSHWs. In contrast, in southern Europe a decrease in soil moisture will cause the region to be in a moisture-depletion state more often, which will decrease land-atmosphere coupling and feedback, reducing temperature variability, and the intensity and range of EuSHWs.

Although we find a robust decrease in EuSHW intensity and their range with global warming in southernmost Europe due to the forced changes in internal variability, and a robust increase in central and northern Europe, models show large differences on the regional patterns and amplitude of the changes which can be generally explained by different projected soil moisture variability changes, in agreement with refs. 52–54. The model differences highlight large uncertainties for regional assessment of the forced changes in internal variability and the need of more SMILEs with high temporal frequency to contrast our results. For example, the latitude at which the contribution of the forced changes in internal variability converts from negative to positive is sensitive to the selection of the climate model. While for CanESM5 and MIROC6 the negative effect is constrained to the southernmost European areas, for ACCESS-ESM-1.5, EC-Earth3 and MPI-GE CMIP6 the negative contribution extends further north covering the Mediterranean coast and the Balkans. It is likely that the model differences in representing the effect of the forced changes in internal variability partly arise from different historical and projected locations of the transitional zone[55], land-surface model parameterization[56], convection schemes[57] or from different soil moisture-vegetation dynamics[58].

We also find regions, mainly over northern Europe, where projected changes in EuSHW intensity due to forced changes in internal variability cannot be explained by soil moisture variability changes. For example, MPI-GE CMIP6 simulates an increase in soil moisture and a decrease in soil moisture variability in northern Europe, conditions that would not explain an increase in EuSHW intensity in that region. This discrepancy between soil moisture changes and EuSHW intensity under rising global mean temperatures suggests that additional factors will play a role in forcing changes in internal variability and consequently affecting future EuSHWs.

One such factor is the disproportionate warming of the Arctic relative to the mid-latitudes[59]. This reduces low-tropospheric meridional temperature gradient, weakens mid-latitude storm tracks, and increases waviness of the jet-stream[60–64]. A weaker storm track and more meandering jet-stream reduce transport of cool marine air into continental regions, increase the frequency of atmospheric blocking, and intensify central EuSHWs[65–67]. In contrast, the tropical upper troposphere is warming at a greater rate compared to the Arctic, strengthening the upper-tropospheric temperature gradient, accelerating and shifting the jet northward, and contributing to a positive trend in the summer North Atlantic Oscillation (SNAO)[68–71]. Since the positive phase of the SNAO is associated with extreme heat events over Northwestern Europe[70,72,73], this trend could explain the increase in northern EuSHW intensity due to forced changes in internal variability shown by MPI-GE CMIP6. However, climate models show substantial disagreement regarding the jet stream response to changing lower-tropospheric and upper-tropospheric meridional temperature gradients[68], which might contribute to uncertainty in projections of forced changes in internal variability and their influence on future heatwave intensity.

Our findings emphasize the role of soil moisture and moisture-temperature feedbacks in modulating the intensity and variability of EuSHWs under increasing global warming levels, with important implications for adaptation strategies. Here we show that while adaptation to increasing mean daily maximum temperatures in southern Europe should be enough to reduce the vulnerability to increasing EuSHW intensity, in central and northern Europe adaptation to increasing mean daily maximum temperatures will not be enough due to the forced changes in internal climate variability. In these regions, forced changes in internal variability will intensify EuSHWs, increasing risks to infrastructure[74,75], agriculture[76,77], and health systems[78,79]. Additionally, increased summer soil moisture variability in central and northern Europe might have implications for fire regimes. Wet years that fuel vegetation growth will be followed more frequently by moisture-limited years leading to a drying and burning of the vegetation[39,80,81]. Nevertheless, the critical role of soil moisture also brings an opportunity as a mitigation strategy for future EuSHWs. In central and northern Europe, the intensification of summer heatwaves driven by forced changes in internal variability could be mitigated by managing soil moisture. Strategies such as improved crop irrigation practices or groundwater conservation policies could help maintain soil moisture levels, suppress land-atmosphere feedback, and reduce both the intensity and variability of future EuSHWs.

## Methods

### Data

We use five single-model initial-condition large ensembles (*SMILEs*): ACCESS-ESM-1.5 (40 members[40]), CanESM5 (50 members[41]), EC-Earth3 (50 members[42]), MIROC6 (50 members[43]), and MPI-GE CMIP6 (50 members[44]). All SMILEs considered here provide daily temporal resolution for the late historical period (1970–2014) and two shared socioeconomic pathway scenarios (SSP2–4.5, and SSP5–8.5) that span until the end of the 21$^{st}$ century. Our main findings are robust with respect to the inclusion of SSP1–1.9, SSP1–2.6, and SSP3–7.0 scenarios. The ensemble members have identical natural and anthropogenic forcing but start from different initial conditions allowing us to robustly identify the effects of forced changes in the shift of temperature distribution and internal climate variability[45–47]. We test the capability of climate models to simulate the intensity of recent EuSHWs by comparing it to the daily gridded land observations E-OBS dataset for the period 1970–2022[82]. All datasets are linearly interpolated to the MPI-GE CMIP6 grid using Climate Data Operator (CDO) command line tool. While the interpolation spatially smooths temperatures in E-OBS, particularly over areas of complex topography, a common grid between datasets is needed to compare the spatially integrated cumulative heat.

We compute two sets (i.e. non-detrended and detrended) of daily maximum 2 m air temperature (T2max) anomalies from 1970 until 2099 to remove the seasonal cycle[83]. The non-detrended T2max anomalies are computed in reference to a centered 15-day running window and 1985–2014 reference period. The 1985–2014 period is chosen to represent the most recent climatology within the historical simulations. The detrended T2max anomalies are computed in reference to a centered 15-day running mean averaged across all ensemble members for each corresponding day of each year and model. For E-OBS, we compute the detrended T2max anomalies in reference to a centered 15-day running mean, averaged across all climate models for each corresponding day of each year. While the non-detrended dataset retains an increasing mean temperature trend (i.e., forced distribution shift) due to global warming, in the detrended dataset the forced distribution shift is removed, isolating changes in the statistics of natural climate oscillations driven by the altered mean state (i.e., forced changes in internal variability).

We use monthly 2 m air temperature to compute area-weighted global mean surface temperature anomalies (GMTA) in reference to the 1985–2014 period. For soil moisture, we use the variable mean total soil moisture content summed over all soil layers (mrso) averaged over boreal summer (June, July, August).

### EuSHW definition and intensity quantification

We apply a percentile-based heatwave definition to identify boreal summer (June, July, August) heatwave days[84]. Percentile-based heatwave definitions consist of a temperature threshold defined by a percentile that represents the perception of extreme temperatures, and in a temporal constraint to capture the persistent nature of heatwaves. For each calendar day and European land grid point (10°W–30°E, 35–70°N), a heatwave day is identified when T2max exceeds the 90$^{th}$ percentile based on a centered 15-day running window and 1985–2014 reference period for at least six consecutive days.

We quantify the intensity of EuSHWs using the cumulative heat metric[1]. The cumulative heat is computed as a seasonal integration of heat exceeding the threshold during heatwave days. In addition, we integrate the cumulative heat over the European domain after weighting each grid point by the cosine of the latitude[22]. Hence, with a single number the cumulative heat accounts for the duration, spatial extent, and amplitude of all EuSHWs that occur in each summer. We refer to non-detrended and detrended EuSHW intensity to the cumulative heat computed with non-detrended T2max and detrended T2max, respectively. We scale the simulated EuSHW intensity with the mean observed EuSHW intensity (i.e., E-OBS 1970–2022).

### Sensitivity of the forced signal and ensemble spread to global warming levels

For each summer the forced signal is given by the ensemble mean, and the range due to internal climate variability is given by the ensemble spread computed as ensemble standard deviation. We linearly regress the forced signal in EuSHW intensity and soil moisture to global mean surface temperature anomalies. Similarly, we regress the ensemble spread in EuSHW intensity and soil moisture to global mean surface temperature anomalies. While linear analysis assumes a consistent relationship across all warming levels, non-linear relationship might exist specially at vey high warming levels. We apply a significance test at the 95% confidence level. We define a robust change when at least three out of the five models agree on the regression coefficient sign, and at least three out of the five models show statistically significant changes.

## Data availability

The data generated in this study have been deposited in the public long-term archive repository of the German Climate Computing Center[85] (DKRZ-LTA; https://www.wdc-climate.de/ui/entry?acronym=DKRZ_LTA_1075_ds00040). The gridded land observations E-OBS dataset is publicly available at Copernicus Climate Data Store (https://cds.climate.copernicus.eu/datasets/insitu-gridded-observations-europe?tab=download). The climate model data are publicly available at Earth System Grid Federation (ESGF) nodes (https://esgf.llnl.gov/nodes.html).

## Code availability

The code used in this study[86] is publicly available in the GitHub repository (https://github.com/gbeobidearsuaga/EuSWH_projection_IV_2025).

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

## Acknowledgements

G.B.A. is supported by the Horizon Europe project EXPECT (Towards an Integrated Capability to Explain and Predict Regional Climate Changes) under Grant Agreement 101137656. L.S.G. received funding from the European Union's Horizon Europe Framework Programme under the Marie Skłodowska-Curie grant agreement No 101064940. A.B and J.B are funded by the Deutsche Forschungsgemeinschaft under Germany's Excellence Strategy - EXC 2037 'CLICCS - Climate, Climatic Change, and Society' - Project Number: 390683824, contribution to the Center for Earth System Research and Sustainability (CEN) of University of Hamburg. D.O. is supported by the Max Planck Society for the Advancement of Science. We acknowledge the Large Ensemble community for providing the climate model simulations used in this study and the German Climate Computing Centre (DKRZ) for the computational resources. We acknowledge financial support from the Open Access Publication Fund of Universität Hamburg.

## Author contributions

G.B.A. conceived the idea. G.B.A. and L.S.G. designed the study. G.B.A. performed all analysis, prepared the figures and wrote the first version of the manuscript. G.B.A., L.S.G., A.B., D.O. and J.B. contributed to the interpretation of the results and the final version of the manuscript.

## Funding

## Competing interests

The authors declare no competing interests.
