## [Transparent Peer Review file · Nature Communications]

Increasing central and northern European summer heatwave intensity due to forced changes in internal variability

Corresponding Author: Dr Goratz Beobide-Arsuaga

Version 0:

Reviewer comments:

Reviewer #1

(Remarks to the Author)

Review of "Increasing central and northern European summer heatwave intensity due to forced internal variability changes"

General comments :

This manuscript uses four single-model large ensembles (ACCESS-ESM1.5, CanESM5, MIROC6, MPI-GE CMIP6) under SSP2-4.5 and SSP5-8.5 to disentangle the contributions of a positive shift in summer temperature distributions ("forced distribution shift") versus forced changes in internal variability on future European summer heatwave (EuSHW) intensity. The key finding is that, under warming, central and northern Europe will experience amplified EuSHW intensity and variability due to moisture-limited land–atmosphere feedbacks, whereas southern Europe may see dampened extremes under more persistent drought. This work offers important insights for regional adaptation strategies. However, significant revisions are required to address omitted literature, clarify methodological details, and bolster the robustness of the conclusions. I recommend Major Revision.

Major comments:

1. The Introduction and Discussion must be expanded to acknowledge and compare with the following pivotal studies, which directly address the relative contributions of external forcing versus internal variability to extreme heatwaves in Northern Hemisphere—spanning Europe, southern China, and western North America (e.g., Zhang et al., 2023; Gong et al., 2024; Ma et al., 2024a, 2024b; Sarah et al., 2024). These works have rigorously attributed major recent heatwave events, quantified the respective roles of internal variability and external forcing, and clarified the underlying mechanisms driving extreme heatwaves. Including these references will enrich the manuscript's contextual framework and provide readers with a comprehensive overview of the latest advances in heatwave attribution and dynamics across the Northern Hemisphere.
2. Lines 104–105: The term "mean observed EuSHW intensity" is not tied to a specific period. Clearly state the exact years and observational dataset (e.g. E-OBS 1950–2022, or 1985–2014) used to compute this baseline. This precision is essential for reproducibility and for interpreting the scaling of future projections.
3. The causal chain—radiative forcing → global mean temperature anomalies → forced signal in EuSHW intensity → changes in internal variability—is too condensed. How greenhouse gas and aerosol forcings drive global mean temperature anomalies. How temperature anomalies translate into ensemble-mean heatwave intensity ("forced signal"). How the same warming alters statistics of internal variability via land–atmosphere and circulation feedbacks. A brief schematic or boxed equation could help clarify these steps.
4. Different models yield varying magnitudes—and sometimes opposite signs—of forced-variability changes across regions. Provide regression slopes of detrended EuSHW intensity versus global mean temperature anomalies. Diagnose the dominant physical drivers (soil moisture feedback or circulation patterns) that underlie those sensitivities. Discuss whether inter-model divergence stems from land-surface parameterizations, convection schemes, or differences in SST variability. A summary table or supplementary figure would greatly enhance transparency.
5. Fig. 4b shows limited regions of great than 3-model agreement on increased detrended EuSHW intensity, raising concerns about reliability. Please discuss how the spatial extent of "non-dashed" regions informs confidence in projections. Could you explain why only these four SMILEs were chosen? For instance, would adding other large-ensemble datasets

(e.g., CESM2 or EC-Earth) expand or shift the regions showing robust agreement? Consider evaluating each model's historical performance in simulating heatwave characteristics—such as by comparing simulated versus observed intensity—and then either applying skill-based weights or presenting performance metrics to justify the focus on regions with strong multi-model agreement.

6. The phrase “the decrease in European summer soil moisture has an opposite effect on soil moisture variability changes in different European regions” is confusing. Please rewrite more clearly, for example: “Under global warming, mean summer soil moisture declines continent-wide (Fig. 5a), but its impact on interannual variability differs by region: central and northern Europe exhibit increasing soil-moisture variability, whereas southern Europe shows a reduction in variability (Fig. 5b).” Make sure to link these contrasting variability responses explicitly to the divergent heatwave outcomes.

7. The authors state throughout the manuscript that future forced changes in internal variability will amplify European heatwaves, but they do not specify which internal-variability factors change significantly under warming to drive these extremes. The paper lacks any discussion of the dynamical mechanisms underpinning extreme heatwaves. The discussion acknowledges North Atlantic SST and broad atmospheric dynamics but omits other important circulation drivers. Please add a paragraph on additional large-scale climate factors known to modulate European heatwaves. For example: Ural blocking events and their projected frequency changes. Arctic sea-ice loss and its teleconnections via the jet stream. North Atlantic Oscillation (NAO) and Scandinavian pattern dynamics under warming. Discuss how changes in these climate modes could interact with forced internal variability to influence future EuSHWs.

8. The phrase “forced internal variability change” is cumbersome and potentially confusing. Consider rephrasing to “forced change in internal variability,” and define your terms clearly in the Introduction or Methods.

References:

- Zhang, X., et al., 2023: Increased impact of heat domes on 2021-like heat extremes in North America under global warming. *Nature Communications*, 14, 1690.
- Gong, H., et al., 2024: Attribution of the August 2022 Extreme Heatwave in Southern China: Role of Dynamical and Thermodynamical Processes. *Bulletin of the American Meteorological Society*, 105, E193-E199.
- Ma, K., et al., 2024: Anthropogenic forcing intensified internally driven concurrent heatwaves in August 2022 across the Northern Hemisphere. *npj Climate and Atmospheric Science*, 7, 290.
- Ma, K., et al., 2024: Attribution of the concurrent extreme heatwaves in Northern Europe and Northeast Asia in July 2018. *Atmospheric Research*, 307, 107506.
- Perkins-Kirkpatrick, S., et al., 2024: Extreme Terrestrial heat in 2023. *Nature Reviews Earth & Environment*, 5, 244-246.

(Remarks on code availability)

Reviewer #2

(Remarks to the Author)

The study effectively leverages single-model initial condition large ensembles (SMILEs) with daily temporal resolution, which is a robust methodological choice to disentangle the effects of forced distribution shifts from changes in internal variability. This addresses a key limitation of previous studies that relied on shorter observational records or coarser model resolutions.

The article clearly identifies soil moisture dynamics and land-atmosphere feedbacks as the primary drivers behind the regional differences in heatwave intensity and variability. The explanation that southern Europe, already moisture-depleted, experiences reduced temperature variability due to consistent aridity, while central/northern Europe's increased soil moisture variability leads to enhanced land-atmosphere feedback, is a strong scientific argument.

The distinction between the nonlinear increase in EuSHW intensity and the linear increase in their intensity range with global warming levels is an important nuance, highlighting not just hotter extremes but also greater unpredictability. However, there are several issues that need to be address before being considered for publication.

1. While the paper defines "Forced Internal Variability Change", a more explicit and concise explanation of what "forced internal variability change" is (e.g., changes in the statistics of natural climate oscillations driven by the altered mean state) early in the introduction or methods would enhance clarity for a broader audience. This would clarify how it differs from unforced, natural internal variability.

2. The text mentions "push the region to switch more often between moisture limited and energy limited states." A brief, more mechanistic explanation of how Soil Moisture Feedback in Central/Northern Europe is switching specifically amplifies land-atmospheric feedback and increases temperature variability would be beneficial (e.g., how the shift between latent and sensible heat fluxes is enhanced).

3. The conclusion briefly touches on adaptation. Expanding the discussion to consider the socio-economic implications of increased variability in central/northern Europe (e.g., challenges for infrastructure, agriculture, health systems, and emergency response beyond just temperature adaptation) could strengthen the real-world relevance.

4. While the paper states the forced distribution shift is the "main contributor," providing a more quantitative comparison of

the relative magnitudes of change attributed to each factor (perhaps a percentage contribution where possible) would offer a clearer picture of their respective importance.

5. If feasible, a conceptual diagram or schematic illustrating the different soil moisture regimes and how they lead to contrasting temperature variability responses in the north/center versus south of Europe could significantly aid understanding.
6. While the chosen SMILEs are well-established, a brief justification for their selection could be added. Are they chosen to represent a range of climate model sensitivities, or for their specific characteristics (e.g., representation of land-atmosphere coupling)?
7. While stating linear interpolation using CDO is fine, mentioning the potential implications of interpolating E-OBS to the MPI-GE CMIP6 grid, especially if there are significant resolution differences, would be useful. Were any other interpolation methods considered, and if so, why was linear chosen?
8. The text mentions computing anomalies to "remove any bias that might arise when computing the intensity of EuSHWs due to the seasonal cycle." However, when comparing model output to E-OBS, direct biases in temperature magnitudes or variability might exist between models and observations. While not the primary focus, a brief comment on how these potential biases are handled or if they are assumed to be less relevant for anomalies and trends would be beneficial.
9. A short rationale for choosing the 1985-2014 reference period for percentile calculations would be useful. This is a common choice, but explaining why this particular period was chosen over others (e.g., for its representativeness or data availability) can strengthen the methodology.
10. For the detrended E-OBS anomalies, it states "in reference to a centered 15-day running window over all ensemble members from all models for each year." Given E-OBS is observational, it doesn't have "ensemble members." It likely refers to averaging over different historical realizations or perhaps the temporal mean. Clarification here would be helpful.
11. For "spatially weighted global mean surface temperature anomalies (GMTA)," specifying the weighting (e.g., area-weighted) would ensure precision, although it's generally assumed for GMTA.
12. The study uses SSP2-4.5 and SSP5-8.5. While these cover a range of plausible futures, it's worth noting that conclusions are bound to these specific scenarios and might not fully capture outcomes under very low emission pathways or other highly uncertain future scenarios. So it is suggested to consider other scenarios that lies in between them.
13. Soil moisture is defined as "fraction of water accumulated in the root zone relative to the water capacity." This is a standard definition, but the precise depth of the root zone and what "water capacity" refers to (e.g., field capacity, total pore space) can vary between models and influence results. A brief acknowledgment of this model-specific variability would be appropriate.
13. While linear regression of heatwave intensity/range to GMTA is a useful approach, it assumes a consistent relationship across all warming levels. While justified in the results (linear for range, non-linear for forced signal), acknowledging that complex, non-linear relationships might be present, especially at very high warming levels, could be a minor limitation.
14. The study focuses on daily maximum 2m air temperature. While this is appropriate for heatwaves, other metrics like duration/return period of heatwaves could also have significant impacts and might be influenced differently by internal variability changes.
15. The paper explicitly states that some observed changes in EuSHW intensity cannot be solely explained by soil moisture variability changes (e.g., CanESM5 north of the Baltic Sea, MPI-GE CMIP6 in northern Europe). This suggests that other factors, such as North Atlantic sea surface temperature variability or atmospheric circulation changes, play an unquantified role, indicating an incomplete understanding of all drivers. Some discussion or explanation (with results if available) is beneficial for the readers.
16. The definition of heatwaves using a fixed percentile (assuming a "time-invariant" perception of extreme temperatures) is a common approach but represents a simplified view of adaptation. Real-world adaptation might involve shifting thresholds as societies acclimatize, which could alter the perceived intensity and impact of future heatwaves. A short analysis with variable percentiles may be beneficial for the study.

(Remarks on code availability)

None

Reviewer #3

(Remarks to the Author)

The intensity and frequency of heatwaves in Europe have significantly increased in recent decades, drawing widespread attention from the academic community. While most studies focus on the impact of anthropogenic forcing on heatwaves, this study investigated the influence of internal variability on heatwave variability in central and northern Europe. They also

indicated the important role of soil moisture in the changes of European heatwave intensity. However, the conclusions drawn in this study exhibit apparent inter-model inconsistencies and lack solid mechanistic analysis, which raises major concerns in my view, as detailed below.

1. This study primarily highlights the importance of internal variability in the long-term changes of European heatwaves. However, as seen in Figure 1, the influence of external forcing remains dominant. It is recommended to include a discussion on the relative impacts of external forcing versus internal variability.

2. As seen in Figure 2, the regions with the highest range of heatwaves caused by internal climate variability are mainly concentrated in southern Europe. However, the range of detrended heatwave intensity resulting from forced changes in internal variability predominantly occurs in central-northern Europe. Further analysis and discussion are required to provide some explanations.

3. The main conclusion of this study is that the increasing heatwave intensity linked to forced changes in internal variability primarily occurs in central-northern Europe, which could result in a faster increase in extreme values compared to the mean. However, this conclusion is largely based on four single-model initial-condition large ensembles, and the results from these four models appear inconsistent (Supplementary Figure 2). Among the four models, the positive contributions of forced changes in internal variability can be found in northern, central, and southern Europe. This inconsistency significantly undermines the reliability of the conclusion, which is my primary concern regarding this paper.

4. The authors attempted to link the effects of forced changes in internal variability with soil moisture changes. However, as analyzed in the paper, soil moisture changes often fail to explain heatwave variability. Moreover, the conclusion that "In central and northern Europe, a decrease in soil moisture will cause the region to switch more frequently between moisture-limited and energy-limited states, amplifying land-atmosphere feedbacks..." lacks in-depth mechanistic analysis. For instance, no supporting evidence is provided to demonstrate that soil moisture actually enhances land-atmosphere coupling in these regions.

5. Why were only four models used? Could the number of models be increased to obtain more reliable conclusions?

(Remarks on code availability)

Version 1:

Reviewer comments:

Reviewer #1

(Remarks to the Author)

The authors have addressed all my concerns, I thus recommend it for publication in current form.

(Remarks on code availability)

Reviewer #2

(Remarks to the Author)

The authors have revised the manuscript carefully. The manuscript can be accepted in the current form.

(Remarks on code availability)

Reviewer #3

(Remarks to the Author)

I appreciate the authors' efforts in revising the manuscript based on the feedback. The paper is now substantially stronger, and most of my comments have been well-addressed. I would be happy to recommend acceptance once the following minor comments are resolved.

1. L186-187: "All models excluding MIROC6 show a significant increase in the forced signal and the range of EuSHW intensity in central Europe", this description is not accurate. In my view, significant increase in ACCESS-ESM-1.5 primarily appears in northern Europe rather than in central Europe.

2. Supplementary Fig. 6: Should be added as a main figure since it presents one of the key conclusions.

3. The distinct modulation of future heatwaves by soil moisture and land-atmosphere feedback across different regions can be supported by a recent publication (Cai et al. 2024: Pronounced spatial disparity of projected heatwave changes linked to heat domes and land-atmosphere coupling).

(Remarks on code availability)

Detailed response to reviewers

Reviewers comments are shown in blue, author responses are shown in black, and the line numbers reference the revised version of the manuscript without track changes.

Reviewer #1:

This manuscript uses four single-model large ensembles (ACCESS-ESM1.5, CanESM5, MIROC6, MPI-GE CMIP6) under SSP2-4.5 and SSP5-8.5 to disentangle the contributions of a positive shift in summer temperature distributions (“forced distribution shift”) versus forced changes in internal variability on future European summer heatwave (EuSHW) intensity. The key finding is that, under warming, central and northern Europe will experience amplified EuSHW intensity and variability due to moisture-limited land–atmosphere feedbacks, whereas southern Europe may see dampened extremes under more persistent drought. This work offers important insights for regional adaptation strategies. However, significant revisions are required to address omitted literature, clarify methodological details, and bolster the robustness of the conclusions. I recommend Major Revision.

We thank the reviewer for their time and for the thoughtful and constructive comments. We have addressed each of the reviewer’s points and revised the manuscript accordingly. Specifically, we have included the missing literature, clarified methodological details, and strengthened the robustness of our conclusions by incorporating an additional model and conducting further analysis. We believe that the reviewer’s comments and suggestions have significantly improved the manuscript.

Major comments:

1. The Introduction and Discussion must be expanded to acknowledge and compare with the following pivotal studies, which directly address the relative contributions of external forcing versus internal variability to extreme heatwaves in Northern Hemisphere—spanning Europe, southern China, and western North America (e.g., Zhang et al., 2023; Gong et al., 2024; Ma et al., 2024a, 2024b; Sarah et al., 2024). These works have rigorously attributed major recent heatwave events, quantified the respective roles of internal variability and external forcing, and clarified the underlying mechanisms driving extreme heatwaves. Including these references will enrich the manuscript’s contextual framework and provide readers with a comprehensive overview of the latest advances in heatwave attribution and dynamics across the Northern Hemisphere.

We thank the reviewer for highlighting these relevant studies. In response, we have added a new paragraph in the Introduction (L49-59), incorporating the suggested literature to discuss the role of internal variability in recent heatwaves.

2. Lines 104–105: The term “mean observed EuSHW intensity” is not tied to a specific period. Clearly state the exact years and observational dataset (e.g. E-OBS 1950–2022, or 1985–2014) used to compute this baseline. This precision is essential for reproducibility and for interpreting the scaling of future projections.

We thank the reviewer for pointing out the missing specification of the time period. We have now clarified this in the Methods section (L417-418) by adding the following sentence: “We scale the simulated EuSHW intensity with the mean observed EuSHW intensity (i.e., E-OBS 1970-2022).”

3. The causal chain—radiative forcing → global mean temperature anomalies → forced signal in

EuSHW intensity → changes in internal variability—is too condensed. How greenhouse gas and aerosol forcings drive global mean temperature anomalies. How temperature anomalies translate into ensemble-mean heatwave intensity (“forced signal”). How the same warming alters statistics of internal variability via land–atmosphere and circulation feedbacks. A brief schematic or boxed equation could help clarify these steps.

We thank the reviewer for the helpful comments, and we agree that the mentioned causal chain in the original manuscript was too condensed. We have expanded the Introduction to provide a more detailed explanation of how global warming could influence EuSHW intensity.

In the second paragraph of the Introduction (L36-47), we describe how global warming is affecting European summer mean temperatures (i.e., forced distribution shift), and how this background warming might influence EuSHW intensity. In the third paragraph (L49-59), we highlight the important role of internal variability in recent and historical EuSHWs. In the fourth paragraph (L61-77), we elaborate on how global warming may also drive changes in internal variability through land-atmosphere and circulation feedbacks, thereby affecting both the intensity and variability of future EuSHWs.

4. Different models yield varying magnitudes—and sometimes opposite signs—of forced-variability changes across regions. Provide regression slopes of detrended EuSHW intensity versus global mean temperature anomalies. Diagnose the dominant physical drivers (soil moisture feedback or circulation patterns) that underlie those sensitivities. Discuss whether inter-model divergence stems from land-surface parameterizations, convection schemes, or differences in SST variability. A summary table or supplementary figure would greatly enhance transparency.

We thank the reviewer for the valuable suggestion.

We provide the regression slopes of detrended EuSHW intensity integrated over Europe versus global mean temperature anomalies, scaled using the mean observed EuSHW intensity (i.e., E-OBS 1970-2022), in L174-179. We believe these scaled regression slopes offer a clearer interpretation than the absolute spatially integrated cumulative heat, which can exceed 1000 °C – values that may initially appear counterintuitive. When spatial integration is not required (i.e., grid-point-level analysis), we present the regression slopes of detrended EuSHW intensity versus global mean temperature anomalies (Fig. 4 and Supplementary Fig. 2).

We carefully considered the suggestion to diagnose the physical drivers (thermodynamic vs. dynamical), but ultimately decided not to pursue this direction. The aim of our manuscript is to investigate how EuSHW intensity is expected to change due to the forced changes in internal variability. Our analysis suggests that changes in soil moisture variability can generally explain the regional patterns in EuSHW intensity changes, consistent with the findings from Vogel et al. (2017) and Maraun et al. (2025). While we agree that a dedicated analysis of thermodynamic and dynamic contributions would be highly valuable, such an analysis would shift the focus of the manuscript and it is out of scope of the present work.

Instead, we have considerably expanded the discussion to consider how the changes in atmospheric circulation might also contribute to changes in EuSHWs (L334-347). Following the reviewer’s suggestion, we also extended the discussion regarding inter-model difference, including possible contributions from land-surface parameterizations, convection schemes, and differing projected atmospheric responses to global warming (L321-324 and L343-347).

5. Fig. 4b shows limited regions of great than 3-model agreement on increased detrended EuSHW intensity, raising concerns about reliability. Please discuss how the spatial extent of “non-dashed” regions informs confidence in projections. Could you explain why only these four SMILEs were chosen? For instance, would adding other large-ensemble datasets (e.g., CESM2 or EC-Earth) expand or shift the regions showing robust agreement? Consider evaluating each model’s historical performance in simulating heatwave characteristics—such as by comparing simulated versus observed intensity—and then either applying skill-based weights or presenting performance metrics to justify the focus on regions with strong multi-model agreement.

We thank the reviewer for suggesting the inclusion of EC-Earth3. We have now incorporated EC-Earth3 into our analysis, which increased the extent of the “non-dashed” areas in Fig. 4 and improved the robustness of our results regarding changes in EuSHW intensity driven by forced changes in internal variability.

Unfortunately, CESM2 provides the required daily-resolution data for only a few number of ensemble members. We identified only one member available at the DKRZ node and four at the NCL node. As shown by Milinski et al. (2020), robust detection of forced changes in internal variability requires approximately 50 ensemble members. Therefore, CESM2 and other SMILEs with considerably fewer than 50 ensemble members at daily resolution are not suitable for inclusion in this study. We have clarified our selection criteria for SMILEs in L87-93.

To assess each model’s performance in simulating the observed spatially integrated EuSHW intensity trend over 1970-2022, we present in Response Fig. 1 the distribution of ensemble members for each model. All models capture the recent increase in EuSHW intensity within their ensemble spread. While most of ensemble members of ACCESS-ESM1-5 and CanESM5 seem to overestimate the trend, the observed trend lies near the center of the ensemble distribution for EC-Earth3, MIROC6, and MPI-GE CMIP6.

We also evaluated each model’s performance at the grid-point level. In Response Fig. 2 we show, for each model and grid-point, the percentile position of the observed EuSHW intensity trend within the ensemble distribution (e.g., red indicates that most ensemble members fall below the observed trend, blue indicates that most ensemble members are above the observed trend, and dashed grid points indicate that the observed EuSHW intensity trend is outside the ensemble spread). We find that, while models seem to underestimate trends in western and eastern Europe and overestimate in northern and southern Europe, all model are able to capture the observed heatwave intensity trend across nearly all grid points.

Given that observations represent only one possible realization of the climate system, and all models capture the observed trends within their ensemble spread, we cannot definitely assess which model performs best. An alternative would be to undertake a process-based evaluation, assessing each model’s performance in simulating the processes relevant for EuSHWs. This would involve evaluating the representation of soil moisture and atmospheric dynamics, ocean-atmosphere interaction and feedbacks, and land-atmosphere interaction and feedbacks, among others. However, evaluating only a subset of these processes could lead to biased or misleading model rankings, as different models could simulate different processes better. A comprehensive process-based evaluation, while highly valuable, is out of scope of the present study.

Response Figure 1: Model evaluation of the EuSHW intensity trend, expressed as spatially integrated cumulative heat trend over the period 1970-2022. The observed trend (E-OBS) is shown alongside the ensemble distribution of simulated trends for: a) ACCESS-ESM1-5; b) CanESM5; c) EC-Earth3; d) MIROC6; e) MPI-GE CMIP6.

Response Figure 2: Model evaluation of the EuSHW intensity trend at the grid-point level, expressed as cumulative heat trend over the period 1970-2022. The color bar indicates the percentile within the ensemble distribution where the observed EuSHW intensity trend lies. Red shades indicate that most ensemble members are below the observed trend, blue shades indicate they are above, and dashed grid points show where the observed trend falls outside the ensemble spread. Models shown: a) ACCESS-ESM1-5; b) CanESM5; c) EC-Earth3; d) MIROC6; e) MPI-GE CMIP6.

6. The phrase “the decrease in European summer soil moisture has an opposite effect on soil moisture variability changes in different European regions” is confusing. Please rewrite more clearly, for example: “Under global warming, mean summer soil moisture declines continent-wide (Fig. 5a), but its impact on interannual variability differs by region: central and northern Europe exhibit increasing soil-moisture variability, whereas southern Europe shows a reduction in variability (Fig. 5b).” Make sure to link these contrasting variability responses explicitly to the divergent heatwave outcomes.

Thank you, we have rephrased the sentence accordingly (L239-242) as follows: “Under global warming, mean summer soil moisture declines continent-wide (Fig. 5a), but its impact on interannual variability differs by region (Fig. 5b). We find a robust increase in soil moisture variability in central and northern Europe, where we also find a robust increase in EuSHW intensity and its range due to the forced changes in internal variability (Fig. 4)...”

7. The authors state throughout the manuscript that future forced changes in internal variability will amplify European heatwaves, but they do not specify which internal-variability factors change significantly under warming to drive these extremes. The paper lacks any discussion of the dynamical mechanisms underpinning extreme heatwaves. The discussion acknowledges North Atlantic SST and broad atmospheric dynamics but omits other important circulation drivers. Please add a paragraph on additional large-scale climate factors known to modulate European heatwaves. For example: Ural blocking events and their projected frequency changes. Arctic sea-ice loss and its teleconnections via the jet stream. North Atlantic Oscillation (NAO) and Scandinavian pattern dynamics under warming. Discuss how changes in these climate modes could interact with forced internal variability to influence future EuSHWs.

We thank the reviewer for the valuable suggestion. We have now expanded both the Introduction and the Discussion sections to incorporate the dynamical mechanisms underlying heat extremes.

In the Introduction (L49-59), we now describe how anomalous sea surface temperatures can modulate atmospheric circulation, and how such circulation anomalies can lead to heatwaves through land-atmosphere feedbacks. Furthermore, in the Discussion (L334-347), we elaborate on how atmosphere dynamics are expected to evolve under global warming, and how these atmospheric changes may contribute to both the forced changes in internal variability and the projected changes in European summer heatwave intensity. We also discuss how these dynamical shifts may contribute to inter-model differences and overall uncertainty.

8. The phrase “forced internal variability change” is cumbersome and potentially confusing. Consider rephrasing to “forced change in internal variability,” and define your terms clearly in the Introduction or Methods.

Thank you for the suggestion. We have rephrased “forced internal variability change” to “forced changes in internal variability” throughout the manuscript, including in the title. Additionally, we have defined more clearly the terms “forced distribution shift” and “forced changes in internal variability” - please see our response to the first comment for more details).

References:

Maraun, D., Schiemann, R., Ossó, A. & Jury, M. (2025). Changes in event soil moisture-temperature coupling can intensify very extreme heat beyond expectations. *Nat. Commun.* **16**, 734.

Vogel, M. M., Orth, R., Cheruy, F., Hagemann, S., Lorenz, R., van den Hurk, B. J., & Seneviratne, S. I. (2017). Regional amplification of projected changes in extreme temperatures strongly controlled by soil moisture-temperature feedbacks. *Geophysical Research Letters*, *44*(3), 1511-1519.

Milinski, S., Maher, N., & Olonscheck, D. (2020). How large does a large ensemble need to be?. *Earth System Dynamics*, *11*(4), 885-901.

Reviewer #2 (Remarks to the Author):

The study effectively leverages single-model initial condition large ensembles (SMILEs) with daily temporal resolution, which is a robust methodological choice to disentangle the effects of forced distribution shifts from changes in internal variability. This addresses a key limitation of previous studies that relied on shorter observational records or coarser model resolutions. The article clearly identifies soil moisture dynamics and land-atmosphere feedbacks as the primary drivers behind the regional differences in heatwave intensity and variability. The explanation that southern Europe, already moisture-depleted, experiences reduced temperature variability due to consistent aridity, while central/northern Europe's increased soil moisture variability leads to enhanced land-atmosphere feedback, is a strong scientific argument.

The distinction between the nonlinear increase in EuSHW intensity and the linear increase in their intensity range with global warming levels is an important nuance, highlighting not just hotter extremes but also greater unpredictability.

However, there are several issues that need to be address before being considered for publication.

We thank the reviewer for their time and for the thoughtful and constructive comments. We have addressed each of the reviewer's points and revised the manuscript accordingly. Specifically, we have conducted three new analyses: (1) we quantified the percentage contribution of forced changes in internal variability; (2) we tested the robustness of our results by including three additional SSP scenarios; and (3) we assessed the impact of using a variable percentile threshold.

In addition, we have expanded the Introduction and Discussion sections to clarify key terms and to incorporate the underlying dynamical and thermodynamical mechanisms driving heat extremes, among other improvements.

We believe that the reviewer's comments and suggestions have significantly strengthened the manuscript.

1. While the paper defines "Forced Internal Variability Change", a more explicit and concise explanation of what "forced internal variability change" is (e.g., changes in the statistics of natural climate oscillations driven by the altered mean state) early in the introduction or methods would enhance clarity for a broader audience. This would clarify how it differs from unforced, natural internal variability.

We thank the reviewer for the valuable suggestion. As a response, we have made a more explicit explanation of what "forced changes in internal variability" are in the Methods section (L392-395) by adding "... in the detrended dataset the forced distribution shift is removed, isolating changes in the statistics of natural climate oscillations driven by the altered mean state (i.e., forced changes in internal variability).".

In addition, we have expanded the Introduction section. In the second paragraph of the Introduction (L36-47), we describe how global warming is affecting European summer mean temperatures (i.e., forced distribution shift), and how this background warming might influence EuSHW intensity. In the third paragraph (L49-59), we highlight the important role of internal variability in recent and historical EuSHWs. We dedicate the fourth paragraph (L61-77) to the forced changes in internal variability. Here, we elaborate on how global warming may also drive changes in internal variability

through land-atmosphere and circulation feedbacks, thereby affecting both the intensity and variability of future EuSHWs.

2. The text mentions "push the region to switch more often between moisture limited and energy limited states." A brief, more mechanistic explanation of how Soil Moisture Feedback in Central/Northern Europe is switching specifically amplifies land-atmospheric feedback and increases temperature variability would be beneficial (e.g., how the shift between latent and sensible heat fluxes is enhanced).

We thank the reviewer for the constructive suggestion. We have now included a mechanistic description of how soil moisture amplifies EuSHWs in transitional regimes by reducing the evaporative cooling (i.e., latent heat flux) and increasing sensible heat flux (55-59). We also clarify that soil moisture-temperature feedback is effective only in transitional regimes. Additionally, we explain that projected future drying in Europe could cause some regions to either enter or exit transitional regimes, thereby increasing or decreasing land-atmosphere coupling and temperature variability (L63-69).

3. The conclusion briefly touches on adaptation. Expanding the discussion to consider the socio-economic implications of increased variability in central/northern Europe (e.g., challenges for infrastructure, agriculture, health systems, and emergency response beyond just temperature adaptation) could strengthen the real-world relevance.

We thank the reviewer for the valuable suggestion. We have now expanded the Discussion section to include the socio-economic implications of increased variability in central and northern Europe (L355-356).

4. While the paper states the forced distribution shift is the "main contributor," providing a more quantitative comparison of the relative magnitudes of change attributed to each factor (perhaps a percentage contribution where possible) would offer a clearer picture of their respective importance.

We thank the reviewer for the valuable suggestion. In response, we have conducted a new analysis to quantify the percentage contribution of forced changes in internal variability to the changes in EuSHW intensity and its range under rising global warming levels. We have included a new figure illustrating these contributions (Supplementary Figure 3) and now report the maximum positive and negative contributions for each model in the manuscript (L189-194 and L208-212).

5. If feasible, a conceptual diagram or schematic illustrating the different soil moisture regimes and how they lead to contrasting temperature variability responses in the north/center versus south of Europe could significantly aid understanding.

We thank the reviewer for the suggestion. Conceptual diagrams illustrating different soil moisture regimes and their implications for land-atmosphere feedbacks have already been published. We refer readers to one such example, shown in Fig. 5 of Seneviratne et al. (2010) (L63-66).

6. While the chosen SMILEs are well-established, a brief justification for their selection could be added. Are they chosen to represent a range of climate model sensitivities, or for their specific characteristics (e.g., representation of land-atmosphere coupling)?

We thank the reviewer for the valuable comment. We have used all available SMILEs that provide the required variables at daily temporal resolution for the SSP2-4.5 and SSP5-8.5 scenarios, each with approximately 50 ensemble members. As demonstrated by Milinski et al. (2020), robust

detection of forced changes in internal variability requires around 50 ensemble members. We now provide a justification for our model selection in L87-93.

7. While stating linear interpolation using CDO is fine, mentioning the potential implications of interpolating E-OBS to the MPI-GE CMIP6 grid, especially if there are significant resolution differences, would be useful. Were any other interpolation methods considered, and if so, why was linear chosen?

We thank the reviewer for the constructive comment. Given that more advanced interpolation methods do not stand out as much more superior than simpler approaches (Hofstra et al., 2008), we choose the commonly used linear interpolation for its relative simplicity and low computational cost. While we acknowledge that interpolation can have implications – particularly for high-resolution datasets over regions with complex topography - a common grid across datasets is necessary to enable comparison of spatially integrated cumulative heat (L380-382).

8. The text mentions computing anomalies to "remove any bias that might arise when computing the intensity of EuSHWs due to the seasonal cycle." However, when comparing model output to E-OBS, direct biases in temperature magnitudes or variability might exist between models and observations. While not the primary focus, a brief comment on how these potential biases are handled or if they are assumed to be less relevant for anomalies and trends would be beneficial.

We thank the reviewer for the valuable comment. We agree that the original sentence "... to remove any bias that might arise ...", was misleading. Following Brunner et al. (2024), we removed the seasonal cycle to eliminate a statistical artifact that arises when applying a running window for threshold calculation. Since we do not apply any bias correction to the temperature magnitude or variability relative to observations, we have revised the sentence to "We compute two sets (i.e. non-detrended and detrended) of daily maximum 2m air temperature (T2max) anomalies from 1970 until 2099 to remove the seasonal cycle" (L384-385).

9. A short rationale for choosing the 1985-2014 reference period for percentile calculations would be useful. This is a common choice, but explaining why this particular period was chosen over others (e.g., for its representativeness or data availability) can strengthen the methodology.

We thank the reviewer for the valuable comment. We have now included the following sentence in L387-388: "The 1985-2014 period is chosen to represent the most recent climatology within the historical simulations."

10. For the detrended E-OBS anomalies, it states "in reference to a centered 15-day running window over all ensemble members from all models for each year." Given E-OBS is observational, it doesn't have "ensemble members." It likely refers to averaging over different historical realizations or perhaps the temporal mean. Clarification here would be helpful.

We thank the reviewer for the valuable comment. We have clarified this in L390-391 by revising the sentence to: "For E-OBS, we compute the detrended T2max anomalies in reference to a centered 15-day running mean, averaged across all climate models for each corresponding day of each year."

11. For "spatially weighted global mean surface temperature anomalies (GMTA)," specifying the weighting (e.g., area-weighted) would ensure precision, although it's generally assumed for GMTA.

We thank the reviewer for this comment. We have now revised the original sentence from “spatially weighted global mean surface temperature anomalies (GMTA)” to “area-weighted global mean surface temperature anomalies (GMTA)” in L397-398.

12. The study uses SSP2-4.5 and SSP5-8.5. While these cover a range of plausible futures, it's worth noting that conclusions are bound to these specific scenarios and might not fully capture outcomes under very low emission pathways or other highly uncertain future scenarios. So it is suggested to consider other scenarios that lies in between them.

We thank the reviewer for the valuable suggestion. To assess the robustness of our results, we tested the sensitivity to scenario choice by including three additional SSP scenarios. We used MPI-GE CMIP6 model, as it consistently provides the necessary data and 50 ensemble members across five SSP scenarios: SSP1-1.9, SSP1-2.6, SSP2-4.5, SSP3-7.0, and SSP5-8.5.

As shown in Response Fig. 3 and 4, the inclusion of the three additional SSP scenarios has minimal impact on changes in non-detrended EuSHW intensity and its range. Specifically, both the forced signal and the ensemble spread at the end of the 21st century consistently decrease from SSP5-8.5 to SSP1-1.9 (Response Fig. 3a). The forced signal and ensemble spread in EuSHW intensity (Response Fig. 3b,c) also show a very similar increase under global warming when compared to results using only two SSP scenarios. For example, the change in ensemble spread is of 2372 °C per degree of global warming, compared to 2340 °C per degree of global warming using five SSP scenarios. The spatial distribution of changes in EuSHW intensity under global warming levels show virtually identical patterns (Response Fig. 4a,b).

Similarly, Response Fig. 5 and 6 demonstrate that these additional SSP scenarios also have little effect on the changes in detrended EuSHW intensity and its range. Again, both the forced signal and ensemble spread at the end of the 21st century show a consistent decrease from SSP5-8.5 to SSP1-1.9 (Response Fig. 5a). The change in the ensemble mean is of 104 °C per degree of global warming using two SSP scenarios, and 105 °C per degree of global warming using five (Response Fig. 5b). Likewise, the change in the ensemble spread is of 119 °C per degree of global warming using two SSP scenarios, and 133 °C per degree of global warming using five (Response Fig. 5c). The spatial distribution of changes in EuSHW intensity under global warming levels remains practically unchanged compared to the results based on only two SSP scenarios (Response Fig. 6a,b).

We therefore conclude that our main findings are robust with respect to the inclusion of SSP1-1.9, SSP1-2.6, and SSP3-7.0, as now stated in L373-374.

Response Figure 3: Changes in European summer heatwave (EuSHW) intensity for MPI-GE CMIP6. a) Observed, historical and projected non-detrended EuSHW intensity expressed as cumulative heat in the left y-axis and scaled it to the observed historical mean (1970-2022) in the right y-axis. Solid lines indicate the ensemble mean, the shading indicates the 5th and 95th percentile, and the right vertical bars indicate the 5th and 95th percentile for the last 20 years of the historical simulation and each shared socioeconomic pathway (SSP1-1.9, SSP1-2.6, SSP2-4.5, SSP3-7.0, and SSP5-8.5) scenarios; b) the forced signal (i.e., ensemble mean) of non-detrended EuSHW cumulative heat scaled to global mean temperature anomalies (GMTA) relative to 1985-2014 for SSP1-1.9, SSP1-2.6, SSP2-4.5, SSP3-7.0, and SSP5-8.5 scenarios (2014-2100); c) the range (i.e., ensemble spread computed as ensemble standard deviation) of non-detrended EuSHW cumulative heat due to internal variability scaled to GMTA relative to 1985-2014 for SSP1-1.9, SSP1-2.6, SSP2-4.5, SSP3-7.0, and SSP5-8.5 scenarios (2014- 2100).

Response Figure 4: Spatial distribution of changes in European summer heatwave (EuSHW) intensity under global warming levels for MPI-GE CMIP6. Regression coefficients for the period 2014- 2100 and for SSP1-1.9, SSP1-2.6, SSP2-4.5, SSP3-7.0, and SSP5-8.5 scenarios between global mean temperature anomalies relative to 1985-2014 and; a) the forced signal (i.e., ensemble mean) of non-detrended EuSHW cumulative heat; b) the range (i.e., ensemble standard deviation) of non-detrended EuSHW cumulative heat due to internal variability. In the non-dashed regions the model shows significant changes at the 95% confidence level.

Response Figure 5: Changes in European summer heatwave (EuSHW) intensity due to forced changes in internal variability for MPI-GE CMIP6. a) Detrended observed, historical and projected EuSHW intensity expressed as cumulative heat in the left y-axis and scaled it to the observed historical mean (1950-2022) in the right y-axis. Solid lines indicate the ensemble mean, the shading indicates the 5th and 95th percentile, and the right vertical bars indicate the 5th and 95th percentile for the last 20 years of the historical simulation and each shared socioeconomic pathway (SSP1-1.9, SSP1-2.6, SSP2-4.5, SSP3-7.0, and SSP5-8.5) scenarios; b) the forced signal (i.e., ensemble mean) of detrended EuSHW cumulative heat scaled to global mean temperature anomalies (GMTA) relative to 1985-2014 for SSP1-1.9, SSP1-2.6, SSP2-4.5, SSP3-7.0, and SSP5-8.5 scenarios (2014-2100); c) the range (i.e., ensemble spread computed as ensemble standard deviation) of detrended EuSHW cumulative heat due to internal variability scaled to GMTA relative to 1985-2014 for SSP1-1.9, SSP1-2.6, SSP2-4.5, SSP3-7.0, and SSP5-8.5 scenarios (2014-2100).

Response Figure 6: Spatial distribution of changes in European summer heatwave (EuSHW) intensity due to forced changes in internal variability under global warming levels for MPI-GE CMIP6. Regression coefficients for the period 2014- 2100 and for SSP1-1.9, SSP1-2.6, SSP2-4.5, SSP3-7.0, and SSP5-8.5 scenarios between global mean temperature anomalies relative to 1985-2014 and; a) the forced signal (i.e., ensemble mean) of detrended EuSHW cumulative heat; b) the range (i.e., ensemble standard deviation) of detrended EuSHW cumulative heat due to internal variability. In the non-dashed regions the model shows significant changes at the 95% confidence level.

13. Soil moisture is defined as "fraction of water accumulated in the root zone relative to the water capacity." This is a standard definition, but the precise depth of the root zone and what "water

capacity" refers to (e.g., field capacity, total pore space) can vary between models and influence results. A brief acknowledgment of this model-specific variability would be appropriate.

We thank the reviewer for this valuable comment. We now only use the *mrso* variable (total soil moisture content summed over all soil layers, and not in the root zone as mistakenly written before). This change was necessary because EC-Earth3 does not provide soil water capacity (*mrsofc*), which prevents us from calculating the fraction of water accumulated. We are aware of the importance of normalizing total soil moisture content by soil water capacity before comparing model results, as models can differ in both their mean values and variability. However, since water capacity is constant over time, the regression analysis is minimally affected, and our results remain consistent with those presented in the previous version. The changes are now reflected L398-400.

13. While linear regression of heatwave intensity/range to GMTA is a useful approach, it assumes a consistent relationship across all warming levels. While justified in the results (linear for range, non-linear for forced signal), acknowledging that complex, non-linear relationships might be present, especially at very high warming levels, could be a minor limitation.

We thank the reviewer for the comment. We have added the following sentence in L425-427: "While linear analysis assumes a consistent relationship across all warming levels, non-linear relationship might exist specially at vey high warming levels."

14. The study focuses on daily maximum 2m air temperature. While this is appropriate for heatwaves, other metrics like duration/return period of heatwaves could also have significant impacts and might be influenced differently by internal variability changes.

We thank the reviewer for this relevant and insightful comment. We agree that alternative heatwave metrics could yield valuable insight. However, the metric used in our study (i.e., the cumulative heat), already integrates key aspects of heatwaves, including duration, magnitude and spatial extent. While examining how the forced changes in internal variability affects individual components of cumulative heat would indeed be interesting, such an analysis would significantly extend the manuscript and lies beyond the scope of our current study. We have therefore chosen not to pursue this additional analysis at this stage.

15. The paper explicitly states that some observed changes in EuSHW intensity cannot be solely explained by soil moisture variability changes (e.g., CanESM5 north of the Baltic Sea, MPI-GE CMIP6 in northern Europe). This suggests that other factors, such as North Atlantic sea surface temperature variability or atmospheric circulation changes, play an unquantified role, indicating an incomplete understanding of all drivers. Some discussion or explanation (with results if available) is beneficial for the readers.

We thank the reviewer for this constructive suggestion. We now expanded both the Introduction and the Discussion sections to incorporate the dynamical mechanisms underlying heat extremes.

In the Introduction (L49-59), we now describe how anomalous sea surface temperatures can modulate atmospheric circulation, and how such circulation anomalies can lead to heatwaves through land-atmosphere feedbacks. Furthermore, in the Discussion (L334-347), we elaborate on how atmosphere dynamics are expected to evolve under global warming, and how these atmospheric changes may contribute to both the forced changes in internal variability and the projected changes in European summer heatwave intensity. We also discuss how these dynamical shifts may contribute to inter-model differences and overall uncertainty.

16. The definition of heatwaves using a fixed percentile (assuming a "time-invariant" perception of extreme temperatures) is a common approach but represents a simplified view of adaptation. Real-world adaptation might involve shifting thresholds as societies acclimatize, which could alter the perceived intensity and impact of future heatwaves. A short analysis with variable percentiles may be beneficial for the study.

We thank the reviewer for this thoughtful suggestion. While our analysis uses a fixed-percentile heatwave definition, which inherently assumes no adaptation to rising temperatures, applying this definition to detrended daily maximum 2m air temperatures does account for adaptation to increasing mean temperatures (i.e. adaptation to shifting temperature distribution). We agree that future societies may also acclimatize to rising extreme temperatures, which could further alter the perceived heatwave intensity and require an analysis using variable percentiles.

We have conducted a new analysis using a variable percentile, the 90th percentile based on a centered 31-year running window, applied to the non-detrended daily maximum 2m air temperatures. This approach assumes adaptation to rising extreme temperatures. The results show generally weaker and statistically non-significant changes in EuSHW intensity at the 95% confidence level across models (Response Fig. 7 and 8), compared to results obtained with the fixed threshold and detrended daily maximum 2m air temperatures. We would like to emphasize that the larger extent of dashed area in Response Fig. 7 is due to more non-significant areas in individual models (as shown in Response Fig. 8), rather than more spatial disagreement in the sign of the regression coefficients.

Weaker changes are expected: if future societies adapt to rising extreme temperatures, the relative impact of future EuSHWs would resemble that of present-day events. Similarly, the lack of statistical significance is also expected. As shown by Milinski et al. (2020), approximately 50 ensemble members are required to robustly quantify changes in internal variability (i.e., changes in the standard deviation of a distribution). Accurately capturing changes above the rising 90th percentile and their effect on EuSHWs would therefore require a much larger ensemble. Thus, the absence of statistical significance does not necessarily imply the absence of actual changes; rather, it reflects an insufficient number of ensemble members to detect them robustly.

Although these results are not robust due to the limited ensemble size, the regional patterns of EuSHW intensity changes (Response Fig. 8) resemble those associated with forced changes in internal variability (Fig. 4): the increase in EuSHW intensity is concentrated in central and northern Europe, while the decrease is observed in southern Europe (Response Fig. 8). This consistency supports our main conclusions. Nonetheless, we have chosen not to include this analysis in the manuscript, as the moving percentile does not fully capture the nature of changes in internal variability.

Response Figure 7: Spatial distribution of changes in European summer heatwave (EuSHW) intensity under global warming levels using a variable heatwave percentile. Multi-model mean of regression coefficients for the period 2014- 2100 and for SSP2-4.5 and SSP5-8.5 scenarios between global mean temperature anomalies relative to 1985-2014 and; a) the forced signal (i.e., ensemble mean) of EuSHW cumulative heat; b) the range (i.e., ensemble spread computed as ensemble standard deviation) of EuSHW cumulative heat due to internal variability. In the non-dashed regions at least three out of the five models agree on the regression coefficient sign, and at least three out of the five models show significant changes at the 95% confidence level.

Response Figure 8: Spatial distribution of changes in European summer heatwave (EuSHW) intensity under global warming levels using a variable heatwave percentile. Regression coefficients for the period 2014- 2100 and for SSP2-4.5 and SSP5-8.5 scenarios between global mean temperature anomalies relative to 1985-2014 and; a) the forced signal (i.e., ensemble mean) of EuSHW cumulative heat for ACCESS-ESM-1.5; b) the range (i.e., ensemble spread computed as ensemble standard deviation) of EuSHW cumulative heat due to internal variability for ACCESS-ESM-1.5; c,d) same as a,b) but for CanESM5; e,f) same as a,b) but for EC-Earth3; g,h) same as a,b) but for MIROC6; i,j) same as a,b) but for MPI-GE CMIP6. The non-dashed regions show significant changes at the 95% confidence level.

References:

- Brunner, L., & Voigt, A. (2024). Pitfalls in diagnosing temperature extremes. *Nature Communications*, 15(1), 2087.
- Hofstra, N., Haylock, M., New, M., Jones, P., & Frei, C. (2008). Comparison of six methods for the interpolation of daily, European climate data. *Journal of Geophysical Research: Atmospheres*, 113(D21).
- Milinski, S., Maher, N., & Olonscheck, D. (2020). How large does a large ensemble need to be?. *Earth System Dynamics*, 11(4), 885-901.
- Seneviratne, S. I., Corti, T., Davin, E. L., Hirschi, M., Jaeger, E. B., Lehner, I., ... & Teuling, A. J. (2010). Investigating soil moisture–climate interactions in a changing climate: A review. *Earth-Science Reviews*, 99(3-4), 125-161.

Reviewer #3 (Remarks to the Author):

The intensity and frequency of heatwaves in Europe have significantly increased in recent decades, drawing widespread attention from the academic community. While most studies focus on the impact of anthropogenic forcing on heatwaves, this study investigated the influence of internal variability on heatwave variability in central and northern Europe. They also indicated the important role of soil moisture in the changes of European heatwave intensity. However, the conclusions drawn in this study exhibit apparent inter-model inconsistencies and lack solid mechanistic analysis, which raises major concerns in my view, as detailed below.

We thank the reviewer for their time and for the thoughtful and constructive comments. We have addressed each of the reviewer's points and revised the manuscript accordingly. Specifically, we have incorporated an additional model to strengthen the robustness of our conclusions, and conducted two new analyses: (1) quantified the percentage contribution of forced changes in internal variability; (2) quantified the changes in land-atmosphere coupling and feedback under global warming, which support the changes in EuSHW intensity driven by forced changes in internal variability.

In addition, we have expanded the Introduction and Discussion sections to clarify the effects of forced distribution shift and forced changes in internal variability on the EuSHW intensity range. We also discuss the potential contribution of atmospheric circulation changes to the projected changes in EuSHW intensity, which may explain why changes in soil moisture variability do not always align with changes in EuSHW intensity.

We believe that the reviewer's comments and suggestions have significantly improved the quality and clarity of the manuscript.

1. This study primarily highlights the importance of internal variability in the long-term changes of European heatwaves. However, as seen in Figure 1, the influence of external forcing remains dominant. It is recommended to include a discussion on the relative impacts of external forcing versus internal variability.

We thank the reviewer for the valuable suggestion. In response, we have conducted a new analysis to quantify the percentage contribution of forced changes in internal variability to the changes in EuSHW intensity and its range under rising global warming levels. We have included a new figure illustrating these contributions (Supplementary Figure 3) and now report the maximum positive and negative contributions for each model in the manuscript (L189-194 and L208-212).

2. As seen in Figure 2, the regions with the highest range of heatwaves caused by internal climate variability are mainly concentrated in southern Europe. However, the range of detrended heatwave intensity resulting from forced changes in internal variability predominantly occurs in central-northern Europe. Further analysis and discussion are required to provide some explanations.

We thank the reviewer for the constructive comment. We would like to clarify that the change in the range of heatwave intensities shown in Fig. 2 are driven by the combined effects of the forced shift in temperature distribution and forced changes in internal variability, with the former being the dominant factor. Notably, an increase in the range of EuSHW intensity could arise solely from the forced shift in temperature distribution. As temperature distributions shift towards higher values, the heatwave threshold -defined as the fixed 90th percentile relative to 1985-2014 reference period-remains constant, assuming limited adaptation to rising temperatures. As a result, exceedances of

this threshold become more frequent, and the potential variability in heatwave intensity increases. Consequently, the contribution of internal variability to heatwave events becomes more pronounced, even if the internal variability itself is not forced. We have clarified this in L150-159 and L282-287.

3. The main conclusion of this study is that the increasing heatwave intensity linked to forced changes in internal variability primarily occurs in central-northern Europe, which could result in a faster increase in extreme values compared to the mean. However, this conclusion is largely based on four single-model initial-condition large ensembles, and the results from these four models appear inconsistent (Supplementary Figure 2). Among the four models, the positive contributions of forced changes in internal variability can be found in northern, central, and southern Europe. This inconsistency significantly undermines the reliability of the conclusion, which is my primary concern regarding this paper.

We thank the reviewer for raising this concern. We have now included an additional SMILE, EC-Earth3, which increases the extent of the non-dashed areas in Fig. 4 and strengthens the robustness of our results regarding changes in EuSHW intensity driven by forced changes in internal variability.

Unfortunately, other SMILEs such as CESM2 provide the required daily-resolution data for only a few number of ensemble members. As shown by Milinski et al. (2020), robust detection of forced changes in internal variability requires approximately 50 ensemble members. Therefore, CESM2 and other SMILEs with considerably fewer than 50 ensemble members at daily resolution are not suitable for inclusion in this study. We have clarified our selection criteria for SMILEs in L87-93.

While the inclusion of EC-Earth3 has expanded the area showing robust EuSHW intensity changes, inter-model inconsistencies naturally remain. Due to data constraints preventing the inclusion of additional SMILEs, we have responded to the reviewer's concern by expanding the Discussion section regarding inter-model differences. This includes potential contributions from different land-surface parameterizations, convection schemes, and differing projected atmospheric responses to global warming, all of which might contribute to divergent regional responses to forced changes in internal variability (L321-324 and L343-347).

More importantly, we would like to highlight that our multi-model mean findings align with the classical conceptual framework in hydrology (Seneviratne et al., 2010), which suggests that soil moisture feedbacks can either be amplified or reduced under declining mean soil moisture, depending on the evapotranspiration regime (L63-69). While the regional assessment on how EuSHW intensity will change due to the forced changes in internal variability (e.g., the latitude at which the increasing EuSHW intensity becomes negative) is very uncertain and model dependent, the fact that a latitude at which EuSHW intensity changes from positive to negative should exist is much more certain. We believe that the agreement between the multi-model mean analysis and classical conceptual framework in hydrology strengthens of our main conclusion.

4. The authors attempted to link the effects of forced changes in internal variability with soil moisture changes. However, as analyzed in the paper, soil moisture changes often fail to explain heatwave variability. Moreover, the conclusion that "In central and northern Europe, a decrease in soil moisture will cause the region to switch more frequently between moisture-limited and energy-limited states, amplifying land-atmosphere feedbacks..." lacks in-depth mechanistic analysis. For

instance, no supporting evidence is provided to demonstrate that soil moisture actually enhances land-atmosphere coupling in these regions.

We thank the reviewer for the valuable comment. We agree that changes in soil moisture variability do not always explain changes in EuSHW intensity due to the forced changes in internal variability, particularly over northern Europe. This point is now clearly stated in L326-328.

While the mean state of the ocean and atmosphere is changing under global warming, their variability may also be altered, potentially enhancing or suppressing the increase in EuSHW intensity driven by the forced distribution shift. One example is the change in meridional temperature gradients in the lower and upper troposphere, which affects mid-latitude atmospheric circulation and, consequently, the intensity of EuSHWs.

Although the simulated mid-latitude atmospheric circulation response to these gradient changes varies across models, and may contribute to the inter-model uncertainties found in our study, recent studies suggest that the summer North Atlantic Oscillation (SNAO) will exhibit a positive trend under global warming (Liu et al, 2025; Mitevski et al., 2025). Since the positive phase of SNAO is associated with northwestern EuSHWs, this trend could help explain our finding that northern EuSHWs are expected to intensity driven by forced changes in internal variability.

We have expanded the Discussion section to address the potential contributions of atmospheric circulation changes to the projected changes in EuSHW intensity, the inter-model differences, and the overall uncertainty (L334-347).

Additionally, we conducted two supplementary analyses to support that the changes in soil moisture variability are associated to changes in land-atmosphere coupling and feedback, and thus to changes in EuSHW intensity.

We compute land-atmosphere coupling as the correlation between the 90th percentile of daily maximum temperature and summer mean soil moisture across ensemble members, and regress this correlation against global mean temperature anomalies. Similarly, we define land-atmosphere feedback as the slope of the linear regression between the same variables, which we also regress against global mean temperature anomalies.

Our results show that under global warming, both the correlation (Response Fig. 9a) and linear regression (Response Fig. 9b) become more positive in southern Europe. Since soil moisture and temperature are typically anti-correlated (i.e., while soil moisture decreases temperature increases), a more positive correlation and regression slope indicate a weakening of land-atmosphere coupling and feedback. In contrast, the correlation and regression slope become more negative in central and northern Europe, indicating a strengthening of land-atmosphere coupling and feedback.

We have included an additional figure to the supplementary material showing changes in land-atmosphere feedback and coupling under increasing global warming levels (Supplementary Figure 6), and have added the relevant text to the manuscript in L245-256.

Response Figure 9: Spatial distribution of changes in land-atmospheric coupling and feedback under global warming levels. Multi-model mean of regression coefficients for the period 2014-2100 and for SSP2-4.5 and SSP5-8.5 scenarios between global mean temperature anomalies relative to 1985-2014 and; a) land-atmospheric coupling computed as the correlation between 90th percentile summer (June, July, August) daily maximum 2m air temperatures and summer mean soil moisture; b) land-atmospheric feedback computed as the linear regression between 90th percentile summer daily maximum 2m air temperatures and summer mean soil moisture. In the non-dashed regions at least three out of the five models agree on the regression coefficient sign, and at least three out of the five models show significant changes at the 95% confidence level.

5. Why were only four models used? Could the number of models be increased to obtain more reliable conclusions?

We thank the reviewer for the constructive comment. We have now incorporated EC-Earth3 into our analysis, which improved the robustness of our results. As stated in the response to the third comment, to our knowledge there are no other SMILEs that fulfill the criteria to robustly identify the forced changes in internal variability.

References:

Liu, Q., Bader, J., Jungclaus, J. H., & Matei, D. (2025). More extreme summertime North Atlantic Oscillation under climate change. *Communications Earth & Environment*, 6(1), 474.

Milinski, S., Maher, N., & Olonscheck, D. (2020). How large does a large ensemble need to be?. *Earth System Dynamics*, 11(4), 885-901.

Mitevski, I., Lee, S. H., Vecchi, G., Orbe, C., & Polvani, L. M. (2025). More positive and less variable North Atlantic Oscillation at high CO₂ forcing. *npj Climate and Atmospheric Science*, 8(1), 171.

Seneviratne, S. I., Corti, T., Davin, E. L., Hirschi, M., Jaeger, E. B., Lehner, I., ... & Teuling, A. J. (2010). Investigating soil moisture–climate interactions in a changing climate: A review. *Earth-Science Reviews*, 99(3-4), 125-161.

Detailed response to reviewers

Reviewers comments are shown in blue, author responses are shown in black, and the line numbers reference the revised version of the manuscript.

Reviewer #1:

The authors have addressed all my concerns, I thus recommend it for publication in current form.

We thank the reviewer for their time. We are pleased to have addressed all of their concerns.

Reviewer #2:

The authors have revised the manuscript carefully. The manuscript can be accepted in the current form.

We thank the reviewer for their time. We are pleased to have addressed all of their concerns.

Reviewer #3:

I appreciate the authors' efforts in revising the manuscript based on the feedback. The paper is now substantially stronger, and most of my comments have been well-addressed. I would be happy to recommend acceptance once the following minor comments are resolved.

We thank the reviewer for their time and for the thoughtful and constructive comments. We have addressed each of the reviewer's points and revised the manuscript accordingly.

1. L186-187: "All models excluding MIROC6 show a significant increase in the forced signal and the range of EuSHW intensity in central Europe", this description is not accurate. In my view, significant increase in ACCESS-ESM-1.5 primarily appears in northern Europe rather than in central Europe.

We thank the reviewer for this comment. We have now modified the sentence to "All models except MIROC6 show a significant increase in the forced signal and the range of EuSHW intensity in central Europe, although in ACCESS-ESM-1.5 this increase is more pronounced in northern than in central Europe." (L186-188).

2. Supplementary Fig. 6 should be added as a main figure since it presents one of the key conclusions.

We thank the reviewer for the suggestion. We have now added the Supplementary Fig. 6 to the main manuscript as Fig. 6.

3. The distinct modulation of future heatwaves by soil moisture and land-atmosphere feedback across different regions can be supported by a recent publication (Cai et al. 2024: Pronounced spatial disparity of projected heatwave changes linked to heat domes and land-atmosphere coupling).

We thank the reviewer for suggesting this relevant publication. We have cited the publication in L310-314.